# Tissue-specific modulation of gene expression in response to lowered insulin signalling in *Drosophila*

Luke Stephen Tain[1†], Robert Sehlke[1,2†], Ralf Leslie Meilenbrock[1], Thomas Leech[1], Jonathan Paulitz[1,2], Manopriya Chokkalingam[2], Nagarjuna Nagaraj[3], Sebastian Grönke[1], Jenny Fröhlich[1], Ilian Atanassov[1], Matthias Mann[3], Andreas Beyer[2,4]*, Linda Partridge[1,5]*

[1]Max-Planck Institute for Biology of Ageing, Cologne, Germany; [2]CECAD Cologne Excellence Cluster on Cellular Stress Responses in Aging Associated Diseases, Cologne, Germany; [3]Department of Proteomics and Signal Transduction, Max-Planck-Institute of Biochemistry, Martinsried, Germany; [4]Center for Molecular Medicine (CMMC) & Cologne School for Computational Biology (CSCB), University of Cologne, Cologne, Germany; [5]Institute of Healthy Ageing, and GEE, UCL, London, United Kingdom

*For correspondence:
andreas.beyer@uni-koeln.de (AB);
partridge@age.mpg.de (LP)

[†]These authors contributed equally to this work

**Competing interests:** The authors declare that no competing interests exist.

**Abstract** Reduced activity of the insulin/IGF signalling network increases health during ageing in multiple species. Diverse and tissue-specific mechanisms drive the health improvement. Here, we performed tissue-specific transcriptional and proteomic profiling of long-lived *Drosophila dilp2-3,5* mutants, and identified tissue-specific regulation of >3600 transcripts and >3700 proteins. Most expression changes were regulated post-transcriptionally in the fat body, and only in mutants infected with the endosymbiotic bacteria, *Wolbachia pipientis*, which increases their lifespan. Bioinformatic analysis identified reduced co-translational ER targeting of secreted and membrane-associated proteins and increased DNA damage/repair response proteins. Accordingly, age-related DNA damage and genome instability were lower in fat body of the mutant, and overexpression of a minichromosome maintenance protein subunit extended lifespan. Proteins involved in carbohydrate metabolism showed altered expression in the mutant intestine, and gut-specific overexpression of a lysosomal mannosidase increased autophagy, gut homeostasis, and lifespan. These processes are candidates for combatting ageing-related decline in other organisms.

## Introduction

Human life expectancy is increasing (*Oeppen and Demography, 2002*) and is predicted to continue to do so (*Kontis et al., 2017*). However, healthspan, the period of life spent in good health and free from the chronic diseases and disorders of ageing, is not keeping up with lifespan and there is therefore a growing period of functional decline and ill health at the end of life (*Crimmins, 2015*; *Niccoli and Partridge, 2012*; *Partridge et al., 2018*). Lowered activity of the insulin and IGF-1-like signalling (IIS) network can extend lifespan in laboratory model organisms (*Kenyon, 2011*; *Partridge et al., 2011*), and possibly humans through specific mutations (*Flachsbart et al., 2009*; *Study of Osteoporotic Fractures et al., 2009*), and can reduce the incidence of age-related impairments and diseases (*Mannick et al., 2014*; *Mannick et al., 2018*). Identifying the molecular mechanisms and understanding exactly how reducing IIS activity prolongs longevity may hence lead to interventions that ameliorate the effects of ageing and prevent age-related pathology.

Gene expression profiling in whole organisms has identified genes and molecular mechanisms that ameliorate ageing in IIS mutants in *C. elegans* (*Ewald et al., 2015*; *Halaschek-Wiener et al.,*

*2005*; *Kaletsky et al., 2016*; *McElwee et al., 2007*; *Murphy et al., 2003*; *Oh et al., 2006*) and *Drosophila* (*Alic et al., 2011*; *Teleman et al., 2008*). Recent transcriptomic analysis in mice (*Page et al., 2018*), and proteomic analysis in *Drosophila* (*Tain et al., 2017*) showed that the responses to lowered IIS are highly tissues-specific. How these tissue-specific responses are regulated is less clear, and tissue-specific profiling of both the transcriptome and the proteome can give a more informative picture, not only of the molecular changes mediating tissue-specific functional responses to mutations that increase healthspan, but also of how gene expression itself is regulated to achieve those responses (*Barrett et al., 2012*).

IIS affects not only lifespan, but also other processes including development, growth, and reproduction (*Bartke, 2011*). Isolating the potentially causal changes in gene expression that specifically modulate longevity in IIS mutants is therefore challenging. In the fruit fly *Drosophila*, IIS is activated through insulin-like peptides (DILPs) (*Grönke et al., 2010*). Genetic ablation of the median secretory neurons (mNSC), which produce DILPs, or null mutation of 3 *dilp* genes (*dilp2-3,5*) that are expressed in the mNSC neurons, systemically lowers IIS, resulting in extended lifespan, reduced body size, reduced female fertility, and delayed development (*Grönke et al., 2010*; *Broughton et al., 2005*). However, the extent to which these traits change is greater in *dilp2-3,5* mutants than mNSC-ablated flies, perhaps because IIS activity is reduced throughout development in *dilp2-3,5* mutants, while lowered IIS commences only later in larval life in mNSC-ablated flies (*Grönke et al., 2010*; *Broughton et al., 2005*).

A naturally occurring endosymbiotic bacterium, *Wolbachia pipientis,* present in many insect species (*Werren and Windsor, 2000*), interacts with IIS (*Ikeya et al., 2009*), and increases the longevity of *dilp2-3,5* mutants (*Grönke et al., 2010*). *Wolbachia* also increases the resistance of *dilp2-3,5* mutants to xenobiotics, but does not affect other phenotypes associated with reduced IIS (*Grönke et al., 2010*). Changes in gene expression that require *Wolbachia* in IIS mutants are therefore potentially causal for longevity, and identifying them could thus aid in isolating the specific mechanisms and processes that mediate IIS mutant longevity.

Here, we have simultaneously profiled tissue-specific changes in gene expression at both the transcriptomic and proteomic level in the gut, brain, thorax, and fat body of *dilp2-3,5* mutant flies. Combining our proteomic analysis with transcriptome profiling, we have examined the role of transcriptional and proteomic responses in remodelling of the tissue-specific proteomes in response to lowered IIS. To pinpoint whether these changes were causal for longevity, we quantified whether these changes in expression were altered in the presence of *Wolbachia*. Surprisingly, we found that, unlike in the gut, brain, and thorax, the majority of fat-body-specific gene expression changes in response to reduced IIS were regulated post-transcriptionally, and that this regulation was entirely dependent upon the presence of *Wolbachia*. To increase the specificity and robustness of our analysis, we performed a novel meta-analysis of the proteomic responses to those in a previously studied mutant, mNSC-ablated flies (*Tain et al., 2017*). Importantly, our tissue-specific gene expression analysis and cross model meta-analysis allowed the identification of both conserved, and model-specific, responses to reduced IIS, which may contribute to IIS-mediated longevity.

We identified novel functional signatures of reduced endoplasmic reticulum (ER)-protein targeting and an increased DNA damage/repair response that require *Wolbachia* and that were specific to the fat body of *dilp2-3,5* mutants. We then showed that DNA damage is reduced, and genome stability increased, in the fat body of *dilp2-3,5* mutant flies and, importantly, that these effects require *Wolbachia*. Furthermore, we showed that increased expression of one subunit of the minichromosome maintenance complex was sufficient to reduce DNA damage in the fat body, and extend lifespan.

Finally, we identified a gut-specific upregulation of lysosomal alpha-mannosidases in response to lowered IIS that occurred only in the presence of *Wolbachia*. Furthermore, we showed that gut-specific overexpression of one lysosomal alpha-mannosidase was sufficient to maintain gut homeostasis and extend lifespan.

## Results

### Tissue-specific remodelling of gene expression in response to reduced IIS

Reducing IIS activity can remodel gene expression via downstream transcription factors (*Partridge et al., 2011*; *Kenyon, 2010*; *Fontana et al., 2010*). To determine the effect of reduced IIS on tissue-specific gene expression, we compared transcript and protein expression levels in *dilp2-3,5* mutants to those of wild-type controls ($w^{Dah}$). We reproducibly identified a total of 11331 transcripts and 7234 proteins (Appendix 1A-C), of which 3683 transcripts and 3738 proteins showed significantly altered expression in at least one tissue of *dilp2-3,5* mutant flies (adj. p-value<=0.1) (Appendix 1D, *Supplementary file 1–2*). In total, the gut, fat body, brain, and thorax of *dilp2-3,5* mutant flies showed 563, 1004, 365, and 2535 differentially expressed transcripts, respectively (Appendix 1D). In contrast, we detected a total of 1824, 1678, and 1473 differentially expressed proteins in the gut, fat body, and brain of *dilp2-3,5* mutant flies, respectively, but only 339 were changed in the thorax (Appendix 1D). Overall both the proteomic and transcriptomic responses to reduced IIS were highly tissue-specific, only 22 proteins and 17 transcripts showed altered expression in all four tissues (Appendix 1D).

### Reduced IIS post-transcriptionally remodels the fat-body-specific proteome

Under steady state conditions, protein abundance is primarily determined by mRNA abundance (*Liu et al., 2016*). To determine if the correlation between mRNA and protein abundance was perturbed in response to lowered IIS activity, we compared tissue-specific transcripts and corresponding proteins in *dilp2-3,5* mutants. On average across the four tissues, two thirds of the significant tissue-specific changes in transcript levels in *dilp2-3,5* mutants were mirrored by changes in expression of the encoded proteins (*Figure 1A*, *Supplementary file 3*). One third, however, were regulated in opposite directions, possibly through post-transcriptional regulation (*Figure 1A*, *Supplementary file 3*).

To more precisely examine the effect of post-transcriptional regulation on the proteome in *dilp2-3,5* mutants, we compared the expression of proteins whose level was significantly regulated in response to reduced IIS, to their associated transcripts. This comparison was made irrespective of whether those transcripts were significantly regulated (*Figure 1B*, *Supplementary file 4*).

Our analysis revealed that gene expression changes in response to reduced IIS changes in the *gut, brain*, and thorax were mainly driven by changes in transcription (*Figure 1B*). However, many gene expression changes in response to lowered IIS in the fat body were post-transcriptional (*Figure 1B*). Over 50% of the proteins changed in the fat body of *dilp2-3,5* mutants were oppositely regulated from their associated transcripts (*Figure 1B*). In total, 1569 proteins were differentially expressed in the fat body of *dilp2-3,5* mutant flies (*Figure 1B*). Of those, 729 proteins changed expression in the same direction as their transcripts and were enriched for functions associated with proteostasis, amino acid metabolism, and mitochondria (*Figure 1B* shown in blue and orange, *Supplementary file 5*). The remaining 840 proteins were regulated in the opposite direction to that of their transcripts. This included 601 significantly downregulated proteins, which were enriched for functions relating to translation/peptide generation, endoplasmic reticulum (ER), and lipids. The remaining 239 proteins were significantly upregulated and enriched for functions relating to DNA replication/repair, and chromatin remodelling (*Figure 1B*, *Supplementary file 5*). Thus, gene expression changes in response to lowered IIS in the brain, gut, and thorax, were driven by transcription; however, in the fat body those changes were primarily post-transcriptional. Together, these findings highlight the importance of quantifying post-transcriptional gene expression when analysing IIS mutants, as many gene expression changes would have been missed if examined solely at the mRNA level.

### Identifying tissue-specific, differential gene expression potentially causal in longevity

To further narrow down changes in gene expression that may have a causal role in IIS mutant longevity, we examined both transcriptomic and proteomic changes in the presence and absence of the

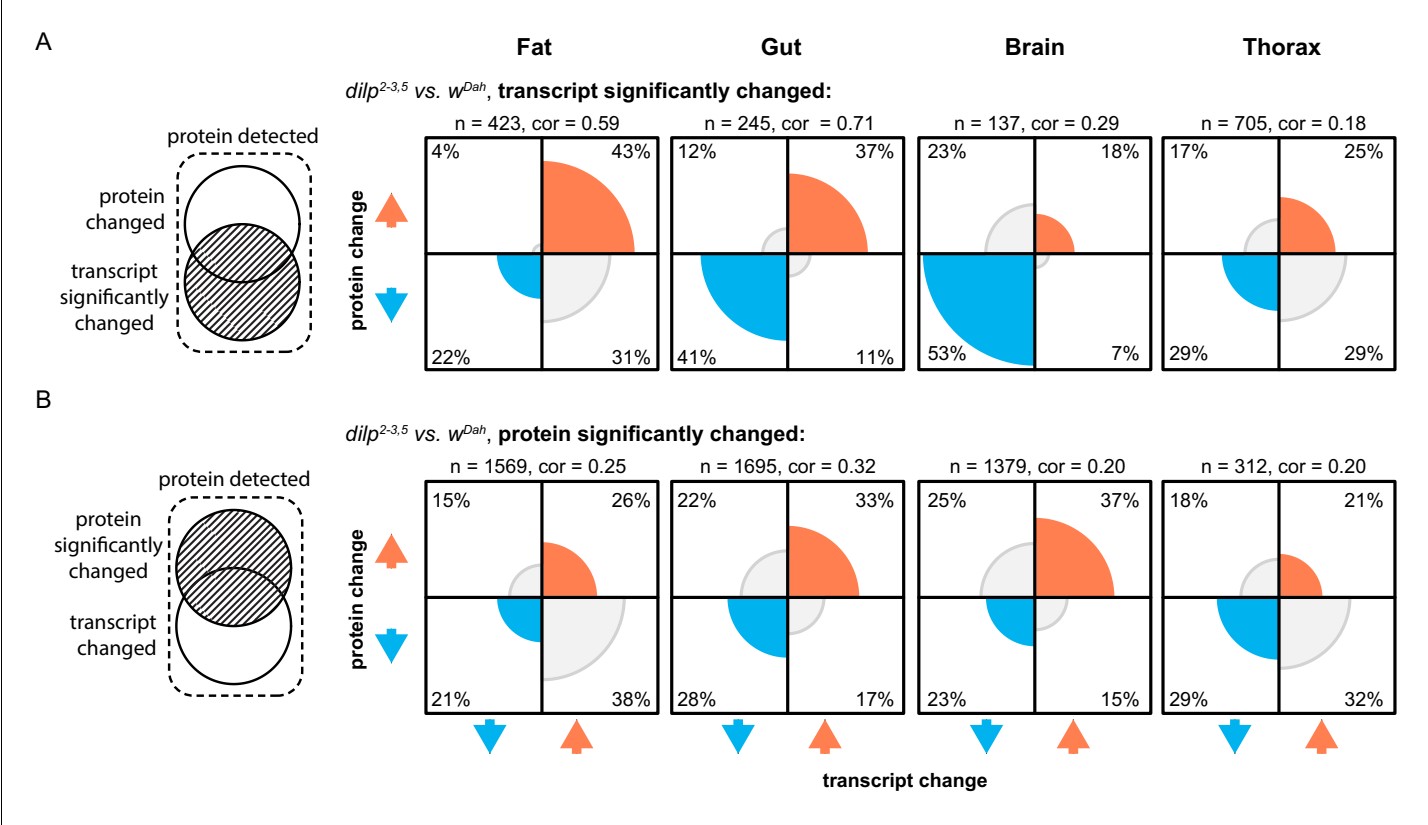

**Figure 1.** Reducing IIS modulates both the tissue-specific transcriptomic and proteomic landscapes. Plots show the proportion of protein/transcript pairs that are regulated in the same (both up [orange] or both down [blue]) or opposite (grey) directions in response to reduced IIS (*dilp2-3,5* vs. *wDah*). Correlations were calculated between the protein and transcript log-fold changes of significantly regulated protein/transcript pairs in each plot. (**A**) All protein/transcript pairs in the respective tissue where the transcript is significantly regulated (adj. p-value≤0.1) in response to reduced IIS, irrespective of if the associated protein is significantly regulated (*Supplementary file 3–4*). (**B**) All protein/transcript pairs in the respective tissue where the protein is significantly regulated (adj. p-value≤0.1) in response to reduced IIS, irrespective of if the associated transcript was significantly regulated (*Supplementary file 3–4*). Correlations (cor.), number of protein/transcript pairs (n) shown above each plot. Rounded percentages of protein/transcript pairs within a specific quadrant of the plots are shown within the respective quadrants (may not total 100%).

intracellular, bacterial symbiont *Wolbachia pipientis*. The presence of *Wolbachia* is required for life-span extension of *dilp2-3,5* mutants, but not for other IIS-related phenotypes such as growth and fecundity (*Grönke et al., 2010*). We identified both transcripts and proteins whose expression changed in the *dilp2-3,5* mutant, but only in the *Wolbachia*-positive background (Appendix 2A), and found both *Wolbachia*-independent and *Wolbachia*-dependent changes (Appendix 2B, *Supplementary file 1–2*). The majority of *Wolbachia*-dependent changes were differentially regulated in the fat body (235 proteins and 249 transcripts) of *dilp2-3,5* mutants, with relatively few changes occurring in the gut, brain, and thorax (Appendix 2B). Of those genes whose *Wolbachia*-dependent expression was altered in the fat body, only 26 were regulated, and regulated in the same direction, on both the transcript and protein level, suggesting considerable post-transcriptional regulation in response to reduced IIS, specifically in this tissue (*Figure 1B*). Surprisingly, we found the previously described post-transcriptional pattern of oppositely regulated proteins and transcripts (*Figure 1B*) was dependent on *Wolbachia* in the fat body of *dilp2-3,5* mutants (Appendix 2D-E).

To identify functional signatures associated to *Wolbachia*-dependent gene expression changes in response to reduced IIS, we performed GO enrichment analysis. Transcripts whose regulation was *Wolbachia*-dependent in fat body were enriched for glucosidase and peptidase enzyme families (Appendix 2C, *Supplementary file 6*). Proteins whose regulation was *Wolbachia*-dependent were enriched for proteins associated with DNA replication and damage/repair responses in fat body, and

mannose metabolism in the gut (Appendix 2C, *Supplementary file 6*). Thus, examining tissue-specific, *Wolbachia*-dependent, changes in gene expression in response to reduced IIS has identified DNA damage/repair responses in fat body and mannose metabolism in the gut as possible regulators of longevity in *dilp2-3,5* mutants. Furthermore, the discrepancy between regulation at the transcript and protein in the fat body suggest increased levels of post-transcriptional regulation in response to reduced IIS.

## Proteomic responses to reduced IIS in two independent *Drosophila* models

Several genetic interventions in *Drosophila* that reduce IIS result in increased lifespan. To identify robust and conserved changes in protein expression in response to reduced IIS, we examined the overlap in differentially expressed proteins between *dilp2-3,5* mutants and the previously published tissue-specific proteomes of mNSC-ablated flies (*Tain et al., 2017*). There was a significant correlation between the tissue-specific changes in each IIS mutant (Appendix 3A-B). However, we also detected 1810 differentially regulated proteins whose expression was not changed in mNSC-ablated flies (Appendix 3B) (*Tain et al., 2017*). Thus, mutant-specific changes occurred, but a significant proportion of the differential expression was also conserved between the two mutants.

Both to increase the power of our analysis and detect shared functional signatures between the two IIS mutants, we performed an undirected network propagation analysis (*Vanunu et al., 2010*), which incorporates protein-protein interaction information. We clustered the resulting network propagation scores (*Supplementary file 7*) that were either shared or detected in only one mutant, and identified functional categories within the clusters with GO enrichment analysis (*Figure 2*, *Supplementary file 8*).

Proteins whose tissue-specific expression pattern was shared between both *dilp2-3,5* mutants and mNSC-ablated flies represent robust and conserved proteomic changes in response to lowered IIS. Those proteins were associated with translation and ribosomal biogenesis, membrane fusion, mitochondrial electron transport, proteostasis, proteasome assembly, and ER protein targeting (clusters 1–8, *Figure 2*). Some of these processes have been directly or indirectly associated with the longevity of IIS mutants (*Tain et al., 2017*; *Essers et al., 2016*; *Augustin et al., 2017*). However, the link between lifespan and fat body-specific ER protein targeting (cluster 8, *Figure 2*) in response to reduced IIS has so far not been explored.

Proteins whose tissue-specific expression pattern was detected in the *dilp2-3,5* mutants, but not mNSC-ablated flies, represent possible mutant-specific proteomic changes in response to lowered IIS. Importantly, as *dilp2-3,5* mutant flies are considerably longer lived than mNSC-ablated flies (*Grönke et al., 2010*; *Broughton et al., 2005*) any *dilp2-3,5*-specific proteomic changes may provide insight into additional mechanisms regulating longevity. DNA damage and repair response was one such functional signature present only in fat body only of *dilp2-3,5* mutants, and may thus represent such an additional pro-longevity response to reduced IIS (Cluster 9, *Figure 2*).

Combining our tissue-specific transcriptomic, proteomic, *Wolbachia*-dependent regulation, and cross model proteomic analyses, of gene expression remodelling in response to reduced IIS led us to examine three main findings that were functional candidates for relevance for longevity. Firstly, we investigated the targeting and translation of proteins to the ER in fat body in more detail. These functional signatures were enriched in the fat body of *dilp2-3,5* mutants and linked to proteins whose levels decreased in response to reduced IIS despite increased expression of the associated transcripts (*Figure 1B*, *Supplementary file 5*). Furthermore, translational and ER-targeting functional signatures were conserved between *dilp2-3,5* mutants and mNSC-ablated flies (*Figure 2*, clusters 7 and 8). Second, we analysed the importance of fat body-specific DNA damage and repair responses, whose functional signatures were identified as both *Wolbachia*-dependent changes in response to reduced IIS and only present in *dilp2-3,5* mutants. Finally, we analysed gut-specific mannose metabolism, which we identified as both gut-specific *Wolbachia*-dependent changes in response to reduced IIS, and conserved between two models of reduced IIS.

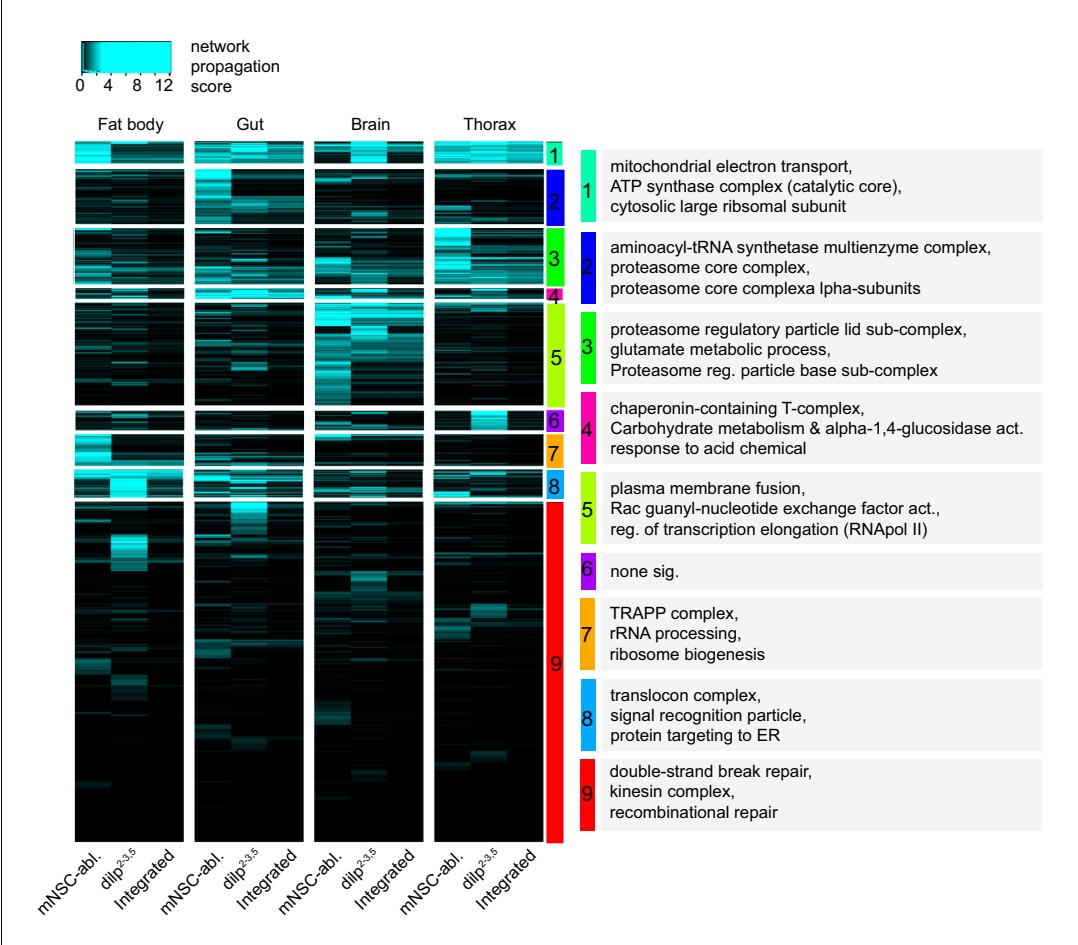

**Figure 2.** Hierarchical clustering and GO enrichment analysis of significantly regulated proteins in two independent models of reduced IIS activity. Tissue-specific heatmap of network propagation scores (*Supplementary file 7*) based on the comparisons of tissue-specific, long-lived IIS mutant proteomes vs. wild-type controls in two independent model systems of reduced IIS: *dilp2-3,5* vs. *w*[Dah] (*dilp2-3,5*) and *InsP3-Gal4/UAS-rpr* vs. *wDah* (mNSC-abl.). For every protein and tissue, the minimum of both scores in that tissue was calculated to show conserved changes between the models (Integrated). Clusters denoted by colour and for each cluster. GO enrichment analysis and selected terms shown in grey boxes (see *Supplementary file 8* for all significant terms).

## Lowered IIS reduces expression of ER-specific co-translational targeting machinery in the fat body

The correct transport and trafficking of newly formed polypeptides within a cell is essential for the creation and maintenance of the distinct subcellular environments required for cellular function. Our analysis identified a fat-body-specific enrichment for proteins associated to the ER and involved in targeting proteins to the ER in *dilp2-3,5* mutants (*Figure 2*, *Supplementary file 8*). To determine if the response to reduced IIS was both tissue- and ER-specific, we calculated average log-fold changes of proteins associated with several cellular compartment terms (*Figure 3A*). ER and Golgi associated proteins were consistently downregulated in the fat body of both mNSC-ablated and *dilp2-3,5* mutant flies (*Figure 3A*). Importantly, this regulation did not appear in other tissues or in the absence of *Wolbachia*, suggesting that reduced ER-targeting of proteins is specific to the fat body and may be causal for the longevity of IIS mutants (*Figure 3A*). Secreted proteins (extracellular space) and intrinsic membrane components, which are processed in the ER, were also downregulated in the fat body in both models of reduced IIS, and only in the presence of *Wolbachia* (*Figure 3A*).

One key mechanism for delivering proteins to the ER is co-translational targeting, the process of importing newly synthesised proteins directly into the ER (*Cross et al., 2009*). Nascent polypeptides

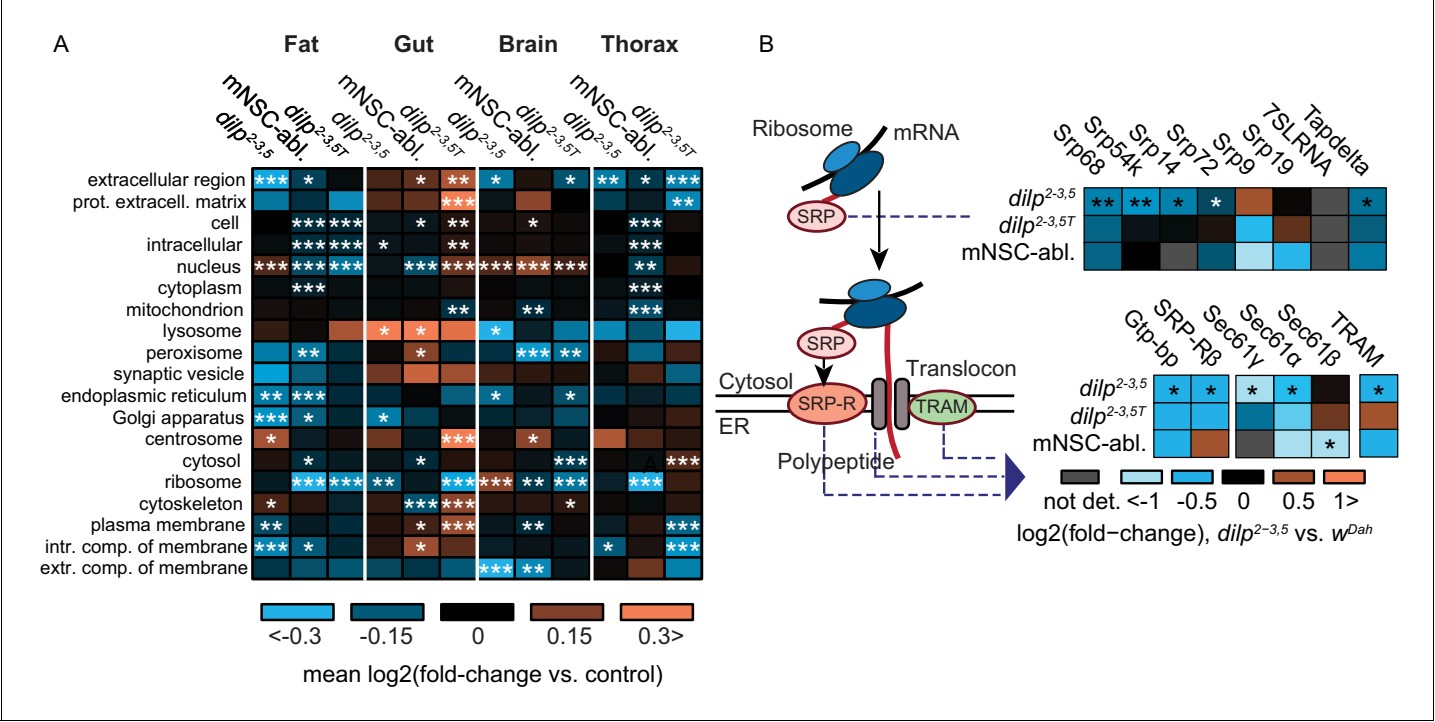

**Figure 3.** Tissue-specific regulation of ER-associated cellular compartments and the ER co-translational targeting machinery in two independent models of reduced IIS. (A) Heatmap of mean log-fold changes in proteins annotated with selected GO cellular compartment terms, in the contrasts *dilp2-3,5* vs. *wDah* (dilp2-3,5), dilp2-3,5T vs. wDahT (dilp2-3,5T (*T* denotes *Wolbachia* minus genotypes, see Methods section) ), and *InsP3-Gal4/UAS-rpr* vs. wDah (mNSC-abl.) flies (*Tain et al., 2017*). Significance of difference vs. zero was calculated using a two-sided Student's t-test (*p<0.05,**p<0.01, ***p≤0.001). (B) Changes in protein expression of SRP, SRP-receptor (SRP-R) sub-units, TRAM and translocon components. Asterisks indicate Benjamini-Hochberg-corrected significance of the limma moderated t-test (*p≤0.1, **p≤0.01, ***p≤0.001).

with signal peptides are recognized by the signal recognition particle (SRP) in the cytosol. SRP-bound nascent peptides are then transported to the SRP-receptor on the ER membrane and passed into the ER lumen through the ER translocon channel, with the aid of translocating proteins (*Saraogi and Shan, 2011*). There, translation is resumed by co-translational targeting through ER-bound ribosomes (*Saraogi and Shan, 2011*). Regulation of the SRP, SRP-receptor, and translocon complex thus determines ER import and co-translational targeting capacity. Tellingly, several components of the ER import and co-translational targeting machinery were down-regulated in the fat bodies of *dilp2-3,5* mutants, and showed a similar trend in mNSC-ablated flies (*Figure 3B*). This included four of the seven SRP subunits (Srp68, Srp54k, Srp14, and Srp72), both SRP-receptor subunits (Gtp-bp and SRPRβ), two of the three translocon core subunits (Sec61gamma and Sec61alpha), and the translocating chain-associating membrane protein (TRAM) (*Figure 3B*). However, although the down-regulation of TRAM and SRP subunits was mostly *Wolbachia*-specific, regulation of the SRP-receptor subunits did not depend on *Wolbachia* (*Figure 3B*). Together, these results suggest that reduced IIS regulates the ER co-translational targeting and protein import machinery, specifically in the fat body of long-lived IIS mutant flies. Furthermore, since much of this regulation was dependent on *Wolbachia*, it may be important for the longevity of IIS mutants.

## DNA damage response and genome stability is increased in the fat body of long-lived IIS mutant flies

A prevalent theory of ageing is that accumulation of molecular damage, including damage to DNA, progressively diminishes cellular function over time and leads to the functional deterioration associated with advancing age (*Maynard et al., 2015*). Several DNA damage and DNA repair pathways exist to prevent and counteract this damage, maintain genomic stability, and in turn maintain cellular and organismal functionality. Our bioinformatic analysis identified a post-transcriptionally increased

abundance of proteins associated with DNA repair and DNA damage responses, in the fat body of *dilp2-3,5* mutants but not in mNSC-ablated flies (*Figure 2*, *Supplementary file 8*). Surprisingly, very few of these proteins were detected as significantly regulated in the fat body of mNSC-ablated flies. This suggests that the increased quality and greater depth of proteomic coverage may have uncovered previously undetected, tissue-specific regulation of these proteins in response to reduced IIS. Most (78%) of the 50 regulated proteins within the GO terms DNA replication and DNA damage/ repair responses were significantly and co-ordinately up-regulated in the fat body of long lived *dilp2-3,5* mutants (*Figure 4A*, *Supplementary file 9*). Furthermore, the regulation of these proteins in *dilp2-3,5* mutants required the presence of *Wolbachia* (*Figure 4A*). Thus, DNA damage/repair responses and genome stability may be increased specifically in the fat body of *dilp2-3,5* mutants, and that increase may be associated with longevity.

In response to DNA damage, including double strand breaks (DSBs), H2AX (His2Av in *Drosophila*) is phosphorylated (Serine 139) (*Rogakou and Sekeri-Pataryas, 1999*). His2Av phosphorylation (p-His2Av) thus serves as an early marker of DNA damage (*Mah et al., 2010*), and can be used to measure DNA damage associated with ageing in model organisms (*Park et al., 2012*; *Wang et al., 2009*) and humans (*Sedelnikova et al., 2008*). To determine if the fat-body-specific upregulation of DNA damage/repair response proteins in IIS mutant flies was sufficient to protect against DNA damage, we quantified the number of p-His2Av foci per nucleus in the fat body of aged (60d) flies (*Figure 4B*). The number of p-His2Av foci was significantly lower in the fat body of aged *dilp2-3,5* mutants compared to similarly aged control flies (wDah) (*Figure 4B*), and only in the presence of *Wolbachia* (*Figure 4B*). Thus, our analysis suggests that DNA damage is reduced in the fat body of IIS mutant flies, and that the reduction in damage may be linked to the longevity.

To determine which proteins may play a role in protecting the fat body from DNA damage, we assessed histone and chromatin proteins. However, with the exception of H2A and H2B in the fat bodies of *dilp2-3,5* mutants we detected no consistent regulation of histones (*Supplementary file 1–2*). In agreement with our previous bioinformatic analysis, proteins associated with the GO-terms chromatin-remodelling, -silencing, -binding, and -organisation were up-regulated in fat body of *dilp2-3,5* mutants (*Supplementary file 9*). For example, the subunits of chromatin-remodelling complexes Tip60 (MRG15, pont, and rept) (*Kusch et al., 2004*), ISWI, Chrac-14, and the INO80 complex were up-regulated in the fat body of IIS mutants flies (*Supplementary file 2*). These complexes each play a role in chromatin remodelling and DNA damage, and may act to maintain genome stability (*Clapier and Cairns, 2009*; *Conaway and Conaway, 2009*). In addition, all six subunits of the replicative helicase minichromosome maintenance complex (MCM2-7) (*Bell and Dutta, 2002*), which is required for both DNA repair and genome stability (*Bailis and Forsburg, 2004*), were up-regulated in response to reduced IIS, and only in the presence of *Wolbachia* (*Figure 4C*). We therefore, tested if tissue-specific increased expression of the minichromosome maintenance complex can decrease the level of DNA damage in the ageing fat body. Of the six up-regulated MCM subunits in the *dilp2-3,5* mutant fat body, MCM6 was up-regulated to the greatest extent (12-fold) (*Supplementary file 2*). Overexpression of MCM6 using the constitutive, fat body-specific, Gal4 driver *Fat body* (*FB*), significantly reduced the number of p-His2Av foci in aged fat body compared to controls (*Figure 4D*). Importantly, fat-body-specific overexpression of MCM6 was sufficient to extend lifespan (*Figure 4E*, *Figure 4—figure supplement 1*). Thus, increased expression of MCM6 in the fat body was sufficient to reduced DNA damage and to extend longevity.

Preventing and repairing DNA damage whilst maintaining chromatin structure and genome stability is essential for cellular and organismal health and their dysfunction is a hallmark of ageing (*López-Otín et al., 2013*; *Wood and Helfand, 2013*). Maintaining the silenced state of heterochromatin in several model organisms, including yeast, worms, flies and, possibly, humans, is important for healthy ageing (*Benayoun et al., 2015*; *Feser and Tyler, 2011*; *Larson et al., 2012*). Loss of heterochromatin structure also occurs during replicative senescence of human fibroblasts in culture, and correlates with increased expression of transposable elements (TEs) (*De Cecco et al., 2013*). Increased expression of TEs with age also occurs in mice (*De Cecco et al., 2013*), flies (*Wood et al., 2016*), and *C. elegans* (*Dennis et al., 2012*). To determine if heterochromatin, and thus genome stability, is maintained in response to reduced IIS activity we analysed the tissue-specific expression of TEs in *dilp2-3,5* mutants and control flies (*w^Dah*). In total, across all tissues, we detected transposon transcripts from 117 of the 176 canonical transposons described for *Drosophila* (*Gramates et al., 2017*; *Figure 4F*, *Supplementary file 10*). Relatively few transposons were significantly regulated in

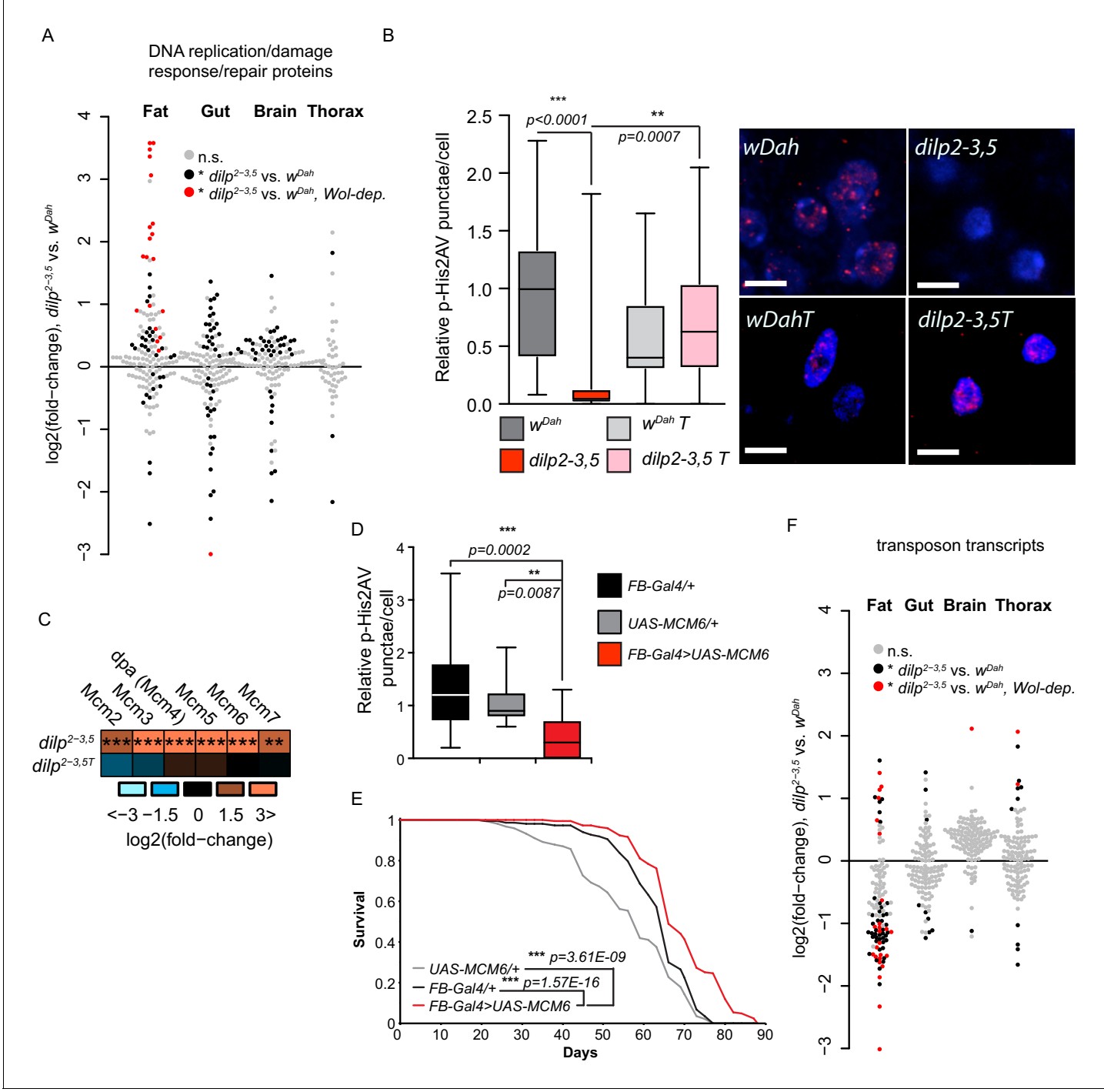

**Figure 4.** Regulation of DNA damage responses and genome stability in response to reduced IIS. (**A**) log2-fold change of DNA replication/DNA damage response proteins, *dilp2-3,5* vs. *wDah* (***Supplementary file 9***). Significantly regulated proteins (black dots), *Wolbachia*-dependent (red dots regulation (adj. p-value≤0.1)), and unregulated (grey dots). (**B**) Relative p-His2Av foci per fat body nuclei in aged (60d) of *dilp2-3,5* mutants compared to controls (*wDah*) in the presence (*dilp2-3,5*) and absence (*wDahT* and *dilp2-3,5T*) of *Wolbachia* (averaged foci/nucleus from independent samples n > 14, scale bar shows 5 μM). (**C**) Regulation MCM complex proteins, *dilp2-3,5 vs. wDah*, in the presence and absence (*dilp2-3,5T*) of *Wolbachia*. Significance of difference vs. zero was calculated using a two-sided Student's t-test (*p<0.05, **p<0.01, ***p≤0.001) exact p values shown in ***Supplementary file 2***. (**D**) Relative p-His2Av foci per fat body nuclei in aged (60d) of flies overexpressing MCM6 specifically in the fat body (*FB-Gal4>UAS-MCM6*, n = 15) compared to genetic controls (*FB-Gal4/+*, n = 16 and *UAS-MCM6/+*, n = 13). Averaged foci/nucleus from independent samples. (**E**) Survival analysis of flies fat-body-specifically overexpressing MCM6 (*FB-Gal4;UAS-MCM6*) compared to the *UAS-MCM6/+* and *FB-Gal4/+* genetic controls. Statistical significance between survival curves was determined by log-rank test (n = 150). (**F**) Differential expression of transposons, *dilp2-3,5* and *wDah*, in each

*Figure 4 continued on next page*

*Figure 4 continued*

tissue (fat, gut, brain, thorax) (**Supplementary file 10**). Significantly changed transposons (black dots), regulated and expression is *Wolbachia*-dependent (red dots) (adj. p-value≤0.1) and not significantly regulated (grey dots).

The online version of this article includes the following figure supplement(s) for figure 4:

**Figure supplement 1.** Replicate lifespans showing tissue-specific expression of UAS-MCM6 or UAS-LManV extends lifespan.

the gut, brain, and thorax, in response to lowered IIS, 9, 2, and 12, respectively (**Figure 4F**). However, 68 transposons were significantly expressed in the fat body of *dilp2-3,5* mutants flies, 12 up- and 56 down-regulated (**Figure 4F**). Of those 68 transposons whose expression was regulated in the fat body of IIS mutant, 24 were only regulated in the presence of *Wolbachia*, and in turn, 75% of those were down-regulated (**Figure 4F**). These findings suggest that reducing IIS activity increases the fat-body-specific maintenance of genome stability, by reducing the expression of TEs, reducing DNA damage, and, together, that these changes may contribute the longevity of IIS mutant flies.

## Lysosomal alpha-mannosidase expression is increased in the gut of long-lived IIS mutant flies and its overexpression is sufficient to extend lifespan

Modulation of gene expression during ageing, or in response to interventions that diminish the effects of ageing, have revealed several conserved pro-longevity metabolic processes (**Murphy et al., 2003**; **Teleman et al., 2008**; **Page et al., 2018**; **Tain et al., 2017**; **Afschar et al., 2016**; **Dobson et al., 2018**; **Hahn et al., 2017**; **Narayan et al., 2016**; **Stout et al., 2013**) including carbohydrate metabolism (**Teleman et al., 2008**; **Afschar et al., 2016**; **Narayan et al., 2016**). Our bioinformatic analysis identified a gut-specific enrichment for proteins involved in carbohydrate metabolism, specifically lysosomal alpha-mannosidases (lysosomal-mannosidases), in both *dilp2-3,5* mutants and mNSC-ablated flies (**Figure 2**). To determine if the response to reduced IIS was tissue-specific, we calculated the average log-fold changes of all significantly regulated lysosomal alpha-mannosidases (**Figure 5A–B**). All lysosomal alpha-mannosidases were detected and significantly upregulated at the protein level, and to a similar extent at the transcript level, in the gut of *dilp2-3,5* mutant flies (**Figure 5A–B**). Importantly, this upregulation did not appear to the same extent in other tissues. Furthermore, the gut-specific upregulation of five out of the six lysosomal alpha-mannosidases only occurred in the presence of *Wolbachia* (**Figure 5B**). Thus, increased levels of lysosomal alpha-mannosidases may be beneficial for longevity.

Gut barrier dysfunction is correlated with reduced lifespan in *Drosophila* and preserving gut homeostasis is correlated with longevity (**Biteau et al., 2010**; **Regan et al., 2016**; **Rera et al., 2012**). We thus tested if increased gut-specific expression of lysosomal mannosidase can maintain homeostasis in the ageing gut. Of the six significantly upregulated lysosomal alpha-mannosidases in the *dilp2-3,5* mutant gut, lysosomal alpha-mannosidase V (LManV) was one of the most significantly regulated (**Figure 5B**). Overexpression of LManV using the constitutive, mid gut-specific, Gal4 driver *Np1* (**Jiang et al., 2009**) increased the level of LManV (**Figure 5—figure supplement 1A**). Flies overexpressing LManV also showed increased numbers of lysosomes in the gut, suggesting increased homeostasis on the cellular level may aid in overall gut function (**Figure 5C**). Gut-specific overexpression of LManV significantly reduced the level of age-related gut barrier failure by 69% compared to controls (**Figure 5D**). Age-related over-proliferation of intestinal stem cells leading to cell crowding, tumour formation, and the resulting loss of gut epithelial organisation may play a major role in gut barrier dysfunction (**Biteau et al., 2010**; **Regan et al., 2016**). Thus, to determine if overexpression of LManV can prevent age-related over-proliferation of intestinal stem cells we quantified the number of phospho-histone-positive (pH3+) cells, a marker for actively dividing cells and a direct measure of stem cell proliferation (**Biteau et al., 2010**). Gut-specific overexpression of LManV significantly reduced the number of pH3+ cells in the gut of aged flies compared to controls (**Figure 5E**). In addition, the reduction in pH3+ cells in flies overexpressing LManV correlated with the amelioration of age-related dysplasia in the gut (**Figure 5F–G**). Finally, gut-specific overexpression of LManV was sufficient to extend lifespan (**Figure 5H**, **Figure 5—figure supplement 1B**). Thus, increased expression of LManV specifically in the gut was sufficient to increase gut homeostasis, maintain gut structure, maintain gut barrier function, and to extend lifespan.

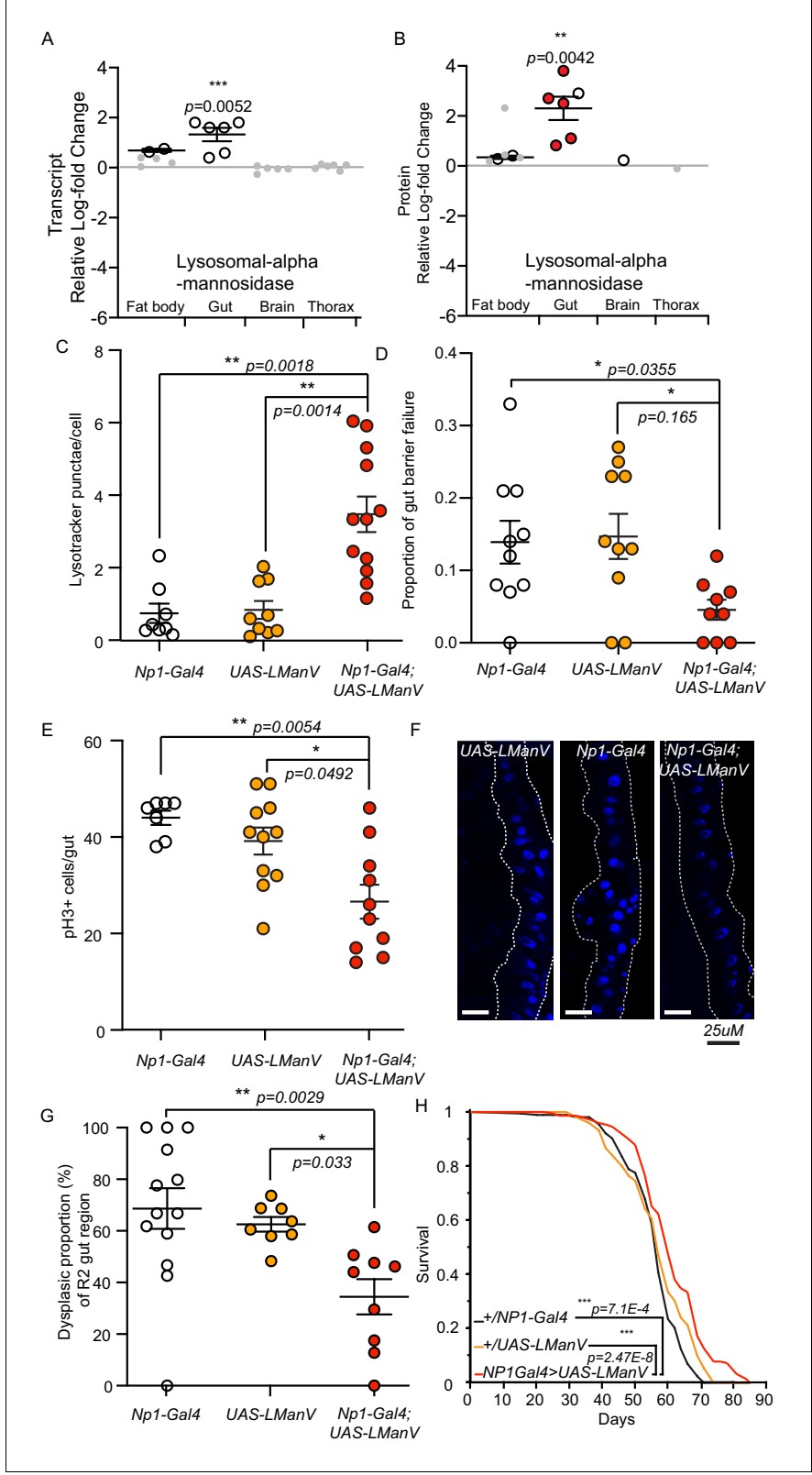

**Figure 5.** Lysosomal alpha-mannosidase expression is gut-specifically increased in response to reduced IIS, and gut-specific expression is sufficient to maintain gut health and extend lifespan. (**A**) Significant (white circles) and non-significant (grey points) tissue-specific log2-fold change of lysosomal alpha-mannosidase transcript (**A**) and protein (**B**) expression in *dilp2-3,5 vs. wDah*. Significantly regulated proteins (white circles), *Wolbachia*-dependent

*Figure 5 continued on next page*

*Figure 5 continued*

(red circles) regulation (adj. p-value≤0.1). Directional significance established by one sided Student's t-test. (C) Quantification of LysoTracker Red stained vacuoles per nucleus in the gut of flies gut-specifically overexpressing LManV (*Np1-Gal4;UAS-LManV, n = 12, Red circles*) compared to genetic controls (*Np1-Gal4/+, n = 8*, white circles and *UAS-LManV/+, n = 9*, orange circles). Chart shows mean and error bars represent S.E.M. Significance determined by Kruskall-Wallace test and Dunn's multiple comparisons test. (D) Proportion of aged (65d) flies exhibiting gut barrier function failure in response to gut-specific overexpression of LManV (*Np1-Gal4;UAS-LManV* red circles) compared to genetic controls (*Np1-Gal4/+* white circles and *UAS-LManV/+* orange circles) (*n = 10*). Significance determined by Kruskal-Wallis test and Dunn's multiple comparisons test. (E) Age-related changes in intestinal stem cell proliferation in response to gut-specific overexpression of LManV (*Np1-Gal4;UAS-LManV, n = 10*, 65d, red circles) compared to genetic controls (*Np1-Gal4/+* white circles, *n = 7*, and *UAS-LManV/+* orange circles, *n = 11*). Significance determined by Kruskal-Wallis test and Dunn's multiple comparisons test. (F) Representative images and (G) quantification of age-related dysplasia in gut epithelia. Gut-specific overexpression of LManV (*Np1-Gal4;UAS-LManV* red circles) significantly reduced age-related intestinal dysplasia in 65-day-old flies compared to genetic controls (*Np1-Gal4/+* white circles and *UAS-LManV/+* orange cirlces). Significance determined by One-way ANOVA and multiple comparisons (Sidak's). (H) Survival analysis of flies gut-specifically overexpressing LManV (*Np1-Gal4;UAS-LManV*, red) compared to the *UAS-LManV/+* (orange) and *Np1-Gal4/+* (black) heterozygous controls. Statistical significance between survival curves was determined by log-rank test (*n = 150*). **<0.01, *<0.05.

The online version of this article includes the following figure supplement(s) for figure 5:

**Figure supplement 1.** Gut-specific overexpression of LManV increases transcript level.

## Discussion

Reduced IIS activity extends lifespan and ameliorates the effects of ageing (*Partridge et al., 2011*; *Fontana et al., 2010*; *Kenyon, 2010*). The diverse roles of IIS (*Bartke, 2011*) and the tissue-specificity of responses to lowered IIS (*Page et al., 2018*; *Tain et al., 2017*) mean the precise molecular mechanisms that mediate IIS mutant longevity remain unclear. In order to identify robust and conserved gene expression changes in response to lowered IIS, we have profiled the tissue-specific transcriptome and proteomes of *dilp2-3,5* mutant flies. Using an interaction between *Wolbachia* and IIS (*Grönke et al., 2010*) and a proteomic comparison to the previously published proteomes of mNSC-ablated flies (*Tain et al., 2017*), we identified expression changes that are both conserved and associated with longevity. We identified fat body-specific post-transcriptional changes in protein expression associated with ER co-translational targeting and DNA damage responses and genome stability that may be causal for longevity. We then quantified a reduction in DNA damage and an increase in genome stability in the fat body of *dilp2-3,5* mutant flies that occurred only in the presence of *Wolbachia*. We then showed that fat body-specific overexpression of MCM6, a DNA damage response protein, was sufficient to reduce DNA damage and extend lifespan. We also identified, transcriptionally driven, gut-specific changes in lysosomal alpha-mannosidases and demonstrated experimentally that gut-specific, but not ubiquitous, ectopic expression of LManV, was sufficient to extend the lifespan of otherwise wild-type flies.

### Profiling tissue-specific gene expression changes in responses to reduced IIS

We profiled expression in four major insulin sensitive tissues, the brain, gut, fat body, and muscle. We have generated a tissue-specific transcript profile of adult *Drosophila* that includes 11331 transcripts. We have also increased the coverage of the adult *Drosophila* proteome by almost 20%, from 6085 (*Tain et al., 2017*) to 7234. In total, we detected the differential expression of 3683 transcripts and 3738 proteins in response to reduced IIS, accounting for 33% and 52% of the detected transcriptome and proteome, respectively. Our analysis thus provides a resource to the scientific community, for both tissue-specific gene expression and also the role of gene expression in IIS-dependent traits.

## Identifying conserved tissue-specific gene expression changes in response to reduced IIS associated to longevity

IIS affects many processes, not only lifespan, including development, growth, and reproduction (*Bartke, 2011*). Thus isolating the specific changes in gene expression that modulate longevity in IIS mutants is problematic. To identify changes in gene expression that may be causal to longevity, we have utilised a known interaction between the endosymbiotic bacteria *Wolbachia* and IIS (*Ikeya et al., 2009*). The interaction between *Wolbachia* and IIS in *dilp2-3,5* mutants results in extended lifespan and increased xenobiotic stress resistance, but does not affect other phenotypes associated with reduced IIS. We identified expression changes, both at the transcript and protein level that only occurred in the presence of *Wolbachia*. Surprisingly, the majority of those changes were detected only at the protein level, and predominantly in a single tissue, the fat body. Thus, our analysis revealed potential tissue-specific and causal regulators of longevity in IIS mutants. Furthermore, we highlight the importance of tissue-specific proteomic, not only transcriptomic, profiling.

To identify robust and conserved changes to the tissue-specific proteomes of long-lived IIS mutant flies, we performed a meta-analysis between the differentially expressed proteins in *dilp2-3,5* mutant flies and the previously published tissue-specific proteomes of mNSC-ablated flies (*Tain et al., 2017*). Along with detecting targets previously unknown to be tissue-specific responses to reduced IIS (described below), our analysis showed regulation of elements known to be key in the responses to reduced IIS in *Drosophila.* These included reduced translation (*Tain et al., 2017*; *Essers et al., 2016*), increased detoxification (*Afschar et al., 2016*), modulation of proteostasis (*Tain et al., 2017*; *Bai et al., 2013*; *Demontis and Perrimon, 2010*; *Tawo et al., 2017*), and mitochondrial respiration (*Tain et al., 2017*). Importantly, despite the independent nature of our two IIS models, and the considerable difference in the strength of the IIS associated phenotypes (size, fecundity, and lifespan), these key responses were shared between the models. We thus conclude that these processes represent robust responses to reduced IIS.

## Fat-body-specific post-transcriptional regulation in IIS mutants requires *Wolbachia*

Gene expression can be controlled both through transcriptional and post-transcriptional regulation (*Liu et al., 2016*). To be able to respond to rapid environmental changes cells can adapt their proteomes through post-transcriptional regulation, as transcriptional regulation may be too slow. To determine how the tissue-specific proteomes of *dilp2-3,5* mutant flies were regulated, we compared them to our tissue-specific transcriptomic profiling of the thorax, gut, brain, and fat body of *dilp2-3,5* mutant flies. The majority of changes in protein expression in the gut, brain, and thorax of *dilp2-3,5* mutants corresponded to similar changes in transcript expression. This was not the case in the mutant fat body, where over 50% of proteins were regulated in the opposite direction to their transcripts. Those genes whose fat body-specific expression was post-transcriptionally regulated in response to reduced IIS were enriched for processes such as DNA damage/repair responses and ER/translation. Furthermore, this regulation required *Wolbachia.* Thus, only profiling the transcriptome would have lead to a misinterpretation of the tissue-specific effects of IIS repression and failed to identify possible mediators of longevity.

## Fat-body-specific ER co-translational targeting is reduced in IIS mutant flies

Declining function of protein homeostasis (proteostasis) leading to reduced cellular viability is one of the hallmarks of ageing (*López-Otín et al., 2013*; *Taylor and Dillin, 2011*). Preventing this decline by reducing translation (*Hansen et al., 2007*; *Pan et al., 2007*; *Wang et al., 2014*), or by increasing expression of proteasomal subunits (*Tain et al., 2017*; *Chen et al., 2006*; *Kruegel et al., 2011*; *Tonoki et al., 2009*; *Vilchez et al., 2012*) can extend lifespan in yeast, *C. elegans* and *Drosophila.* Here, we show that another aspect of proteostasis, protein targeting/trafficking may also play a role in ageing. Sorting proteins to the correct cellular location is a vitally important part of proteostasis and dysfunction in the sorting of proteins can lead to disease (*Balch et al., 2008*). We identified a fat-body-specific downregulation of proteins associated to the ER, and proteins involved in co-translational import into the ER in response to reduced IIS. Importantly, these changes were common to

both *dilp2-3,5* mutants and mNSC-ablated flies, and only occurred in the presence of *Wolbachia*, suggesting this regulation may play a specific role in IIS mutant longevity.

Newly synthesised proteins can be directly trafficked into the ER through co-translational import, via the SRP and translocon (*Saraogi and Shan, 2011*). We quantified several SRP, SRP-receptor, and translocon subunits as significantly down-regulated in the fat body of long-lived IIS mutants. Thus, co-translational import into the ER may be tissue-specifically decreased in response to reduced IIS, possibly as a means of post-transcriptionally regulating the proteome.

Interestingly, the role of ER itself in post-transcriptional regulation is vital, with the ER acting as a major cellular compartment for translation, primarily of secreted and membrane proteins (*Reid and Nicchitta, 2012*; *Reid and Nicchitta, 2015*). The reduction in co-translational import to the ER detected in IIS mutants may therefore result in a specific reduction in translation of secreted proteins. However, our analysis suggests the response is more likely to represent a general reduction in translation, because the expression of both secreted, and membrane, proteins was reduced to the same extent.

Reduced translation is a common feature of long-lived models, especially models of reduced IIS activity, where both translational capacity and translational rate are decreased (*Tain et al., 2017*; *Essers et al., 2016*). Translational capacity is decreased in both the *Drosophila* gut and fat body of IIS mutants, with fewer cytoplasmic ribosomes associating with mRNAs undergoing active translation (*Essers et al., 2016*). Translational rate, however, is only decreased in the fat body of IIS mutant flies (*Tain et al., 2017*; *Essers et al., 2016*). Unlike tissue-specific reductions in translational capacity, fat-body-specific reductions in translational rate are dependent on dFoxo in IIS mutants and are thus potentially important for longevity. Functional signatures associated to a general reduction in translation were detected in both the gut and the fat body. However, changes in the expression of co-translational import machinery proteins were unique to the fat body in response to reduced IIS. Changes in the ER co-translational import machinery proteins could lead to more global changes in translation.

## Fat-body-specific regulation of the DNA damage/repair machinery and maintenance of genome stability in response to lowered IIS

Our study examined the role of DNA damage responses and genome stability in response to reduced IIS. Key elements of the DNA damage response were strongly, and specifically, induced in the fat body of *dilp2-3,5* flies. Furthermore, we found that this response required *Wolbachia* and therefore the proteins of the DNA damage response and genome stability machinery are good candidates for regulators of IIS mutant longevity.

Progressive age-related incidents of DNA damage are common amongst most species examined, and may be the cause cellular dysfunction and thus ageing (*Maynard et al., 2015*). Preventing DNA damage may therefore be beneficial for longevity. Using phosphorylated His2Av as a marker for early DNA damage (*Mah et al., 2010*), we were able to confirm a reduction of DNA damage in the fat body of IIS mutant flies that required the presence of *Wolbachia*. This correlated with the changes in expression of the DNA damage response and repair pathways detected in the fat body of *dilp2-3,5* mutants. We identified the replicative helicase MCM complex as regulated in response to reduced IIS. Furthermore, all subunits of the MCM complex were upregulated, but only in the presence of *Wolbachia*. Thus, the MCM complex may play a role in modulating the longevity of IIS mutant flies.

The MCM complex is an evolutionarily conserved replicative helicase required for initiation and elongation during DNA replication (*Bell and Dutta, 2002*). In addition, the MCM complex has been shown to play a role in transcription, chromatin remodelling, and genome stability (*Bailis and Forsburg, 2004*; *Forsburg, 2004*). More recently, MCM complex proteins have been shown to be recruited to sites of DNA damage (*Drissi et al., 2015*) and play a role in repair responses (*Drissi et al., 2018*). Increased MCM protein levels in the fat body of IIS mutants may aid in preventing age-related DNA damage and/or increase the efficiency of repair pathways and thus promote longevity. Indeed, increased expression of a single subunit (MCM6) of the MCM complex in the fat body was sufficient to significantly reduce age-associated DNA damage in that tissue. Furthermore, increased expression of MCM6 in the fat body was sufficient to extend longevity. Together, these data suggest a role for increased DNA damage/repair responses as possible modulators of IIS-mediated longevity.

These changes also extended to proteins that regulate genome stability. DNA damage and genomic instability are closely linked (*Celeste et al., 2002*). Loss of genome stability leads to deregulation of transcription and has been associated with ageing across a wide range of model organisms including *Drosophila* (*Maxwell et al., 2011*; *Wood et al., 2010*). Combined with the dysregulation of DNA surveillance mechanisms with age, loss of stability can lead to the deleterious activation and mobilisation of TEs (*Dennis et al., 2012*; *Maxwell et al., 2011*; *Chen et al., 2016*). However, interventions exist that can prevent TE activation. Dietary restriction can ameliorate transposon derepression and loss of gene silencing with age (*Wood et al., 2016*; *Jiang et al., 2013*). We find that transposon expression was diminished, specifically in the fat body, in response to reduced IIS. Our study suggests that a direct intervention to reduce IIS in *Drosophila* can tissue-specifically activate DNA maintenance mechanisms to maintain transposon repression and potentially delay genomic instability. One important distinction to be studied in the future will be to determine whether increased genome stability prevents DNA damage, or if reduced DNA damage increases genome stability.

## Gut-specific regulation of lysosomal alpha-manosidase ameliorates age-related gut pathology and is sufficient to extend lifespan

Managing nutrient uptake and regulating metabolism, including carbohydrate metabolism, is vitally important for an organism. Changes to carboydrate metabolism has been linked to ageing and interventions that ameliorate the effects of ageing (*Afschar et al., 2016*; *Narayan et al., 2016*). We identified proteins of a key enzyme family as regulated in response to reduced IIS, namely lysosomal alpha-mannosidases. All the lysosomal alpha-mannosidases were upregulated, but most only in the presence of *Wolbachia.* This suggests lysosomal alpha-mannosidases may play a role in modulating the longevity of IIS mutant flies.

Mutations that affect mannose metabolism enzyme activity result in multiple genetic disorders, with more severe mutations leading to multi-organ insufficiencies (*Freeze et al., 2014*). The normal function of lysosomal alpha-mannosidases is to cleave mannose residues from the oligosaccharides of glycosylated proteins, thereby aiding in lysosomal recycling of glycoproteins (*Nemčovičová et al., 2013*). The question therefore remains, how can increased lysosomal alpha-manosidase expression in the gut extend lifespan? Increased lysosomal alpha-manosidase expression in response to reduced IIS may regulate proteostasis through lysosomal protein recycling. Importantly, our analysis showed increased lysosomal alpha-manosidase expression increased the number of lysosomes in the gut, which may increase gut health on a cellular level. Furthermore, we showed that increased lysosomal alpha-manosidase expression maintained gut health, reduced age-related gut pathologies, and increased longevity. In the future, targeting lysosomal alpha-mannosidases in the gut may therefore prove to be therapeutic against the effects of ageing.

In summary, here we present a highly reproducible, tissue-specific, transcriptomic and proteomic dataset containing over 11,000 transcripts and 7000 proteins quantified from adult *Drosophila* and show how their expression changes in response to reduced IIS activity. To identify robust modulators of longevity, we have examined proteomic changes that are conserved between two models of reduced IIS and utilised the requirement of *Wolbachia* for IIS-mediated longevity to triage those proteomic changes most likely causal for longevity. We show that post-transcriptional regulation plays a major role in remodelling the fat body-specific proteome in response to reduced IIS, and that many possible candidates for mediating longevity are post-transcriptionally regulated in that tissue. We then identified two processes, ER co-translational ER import and DNA-damage responses/genome stability, regulated in the fat body of long-lived IIS mutant flies, potentially causal to their longevity. We also show that fat-body-specific overexpression of MCM6 is sufficient to reduced age-associated DNA damage and extend lifespan. Furthermore, we show that gut-specific overexpression of LManV was sufficient to increase gut homeostasis and extend lifespan. Together, our analysis offers a valuable resource to the scientific community, and highlights the importance of tissue-specific proteomic profiling and cross model comparison to discover potentially unidentified modulators of longevity.

## Materials and methods

### Fly stocks, fly husbandry, and dissection

All flies were backcrossed into a white Dahomey (wDah) strain genetic background for at least eight generations. Naturally Dahomey carry the intracellular bacterium *Wolbachia pipientis*. Wolbachia minus (*wDahT*) flies were generated by treating *wDah* flies with Tetracycline (25 mg/ml in standard SYA food) for three generations followed by a minimum of five generations to allow for full recovery from tetracycline treatment and restoration of intestinal flora. Unless specifically stated all strains, including *FB-Gal4, NP1-Gal4, UAS-MCM6* (RRID:FlyBase_FBst0500029), and *UAS-LManV*, were backcrossed into a *Wolbachia*-positive white Dahomey (*wDah*) strain genetic background. The presence or absence of *Wolbachia* was tested in all stocks via PCR using primers for the *Wolabchia surface protein* (*wsp*), specifically the primer combination wsp81F (AAAAATTAAACGCTACTCCA) and wsp691R (TGGTCCAATAAGTGATGAAGAAAC). *dilp2-3,5T* mutants were generated by back crossing flies into the wDahT background for a minimum of 10 generations (Grönke et al., 2010).

Fly stocks were maintained at 25°C on a 12 hr light and 12 hr dark cycle and fed a standard sugar/yeast/agar diet (Bass et al., 2007). All experimental flies were once mated females, and raised at controlled larval densities. Adult flies were kept in SYA food vials (25 flies per vial) and aged 10 d prior to tissue dissection in cold phosphate buffered saline (PBS) and directly frozen on dry ice. Dissection of *Drosophila* gut included malpighian tubules. Dissected thorax contained predominantly muscle. To generate UAS-LManV flies the ORF was PCR amplified with primers 1 (ATGAAGTTCCTGGGCAT) and 2 (TTATTCATGTTTAATGATGAATGTTCG) and the LManV cDNA UFO03585 (*Drosophila* Genomics Resource Center) and subsequently cloned into the pUAST attb vector (RRID: Addgene_18944). The CDS was amplified and inserted via restriction digest using KpnI and NotI. pUAST attb LManV was inserted into the fly genome by the φC31 and attP/attB integration system (Bischof et al., 2007) using the attP40 landing site.

### Immunohistochemistry

Tissues were dissected in cold PBS and immediately fixed in 4% formahldehyde for 30 min, washed three times in PBS and incubated (1 hr) in blocking solution (0.5% BSA PBS-Tween 0.5%). Tissues were then incubated with anti p-His2Av (Ser137)(1:100)(Rockland immunochemicals Inc, RRID:AB_828383) in blocking solution on a rotator at o/n at 4°C, washed three times, and incubated with anti-rabbit HRP-conjugated secondary antibody (1:1000). Tissues were then washed three times in PBS and mounted in mounting medium (Vectashield H-1200) containing DAPI. Images were taken using confocal microscope (Leica TCS SP5X) with 40 × 1.25 oil objective. Laser power and optical settings were kept constant between images. Images were then analyzed using Imaris 8.0 software (RRID: SCR_007370). The number of p-His2av (Ser137) puncti and DAPI stained nuclei were quantified according to user guidelines and puncti/nucleus calculated. In total between 14 and 17 independent biological replicates were used for quantification (*wDah n = 16, wDahT n = 15, dilp2-3,5 n = 17, dilp2-3,5T n = 14*). Per individual one micrograph was included for quantification each containing a minimum of six fat body cells (based on DAPI-stained nuclei count).

All image analysis and quantification were performed under blinded conditions.

### Intestinal health assays

#### Intestinal dysplasia

Aged (65d) guts from female flies were dissected in cold PBS and fixed in 4% formaldehyde for 30 min, washed, and mounted in mounting medium containing 1.5 ug/ml DAPI (Vetashield, H1200). DAPI was images using a confocal microscope (Leica SP8-X). For each genotype 8–13 guts were imaged in the area proximal to the proventriculus (R2 region). Dysplasia was measured under blinded conditions using ImageJ, and the average proportion of the dysplasic gut relative to the length of gut within the image was calculated. Scoring of dysplasia was performed by measuring the length of gut with several layers of nuclei that formed small nuclear 'nests' of unpolarised cells that clustered atypically and lead to loss of epithelial organisation. Healthy, non-pathological, gut was measured as large absorptive cells aligned to forma single layer epithelium with even spacing.

## Intestinal barrier function

Aged (65d) flies were transferred to SYA food vials containing 2.5% (w/v) Brilliant Blue (AppliChem), a non-absorbable blue dye. A minimum of 160 flies per genotype was assayed. After 48 hr, flies were then examined for the presence of dye leakage into the body as previously described (*Rera et al., 2012*).

## Intestinal stem cell proliferation

Aged (65d) guts from female flies were dissected in cold PBS and fixed in 4% formaldehyde for 30 min, washed three times in PBS and incubated (1 hr) in blocking solution (0.5% BSA PBS-Tween 0.5%). Tissues were then incubated with anti-PH3 (Cell Signalling Technologies 9701 1:500, RRID: AB_331535) in blocking solution on a rotator at o/n at 4°C, washed three times, and incubated with a Alexa Fluor 594 donkey anti-rabbit secondary antibody (Thermo Fisher A21207, 1:1000, RRID:AB_141637). Tissues were then washed three times in PBS and mounted in mounting medium (Vectashield H-1200) containing DAPI. PH3-positive cells were then counted per gut.

## LysoTracker staining and quantification

For LysoTracker staining, fat bodies from female flies (10d) were dissected in PBS and immediately stained with LysoTracker Red DND-99 (Invitrogen) dye (1 µM in PBS) for 2 min. Tissues were then washed three times in PBS and mounted in mounting medium (Vectashield H-1200) containing DAPI. Images were taken using confocal microscope (Leica TCS SP8X) with 40 × 1.25 oil objective. Laser power and optical settings were kept constant between images. Images were then analyzed using Imaris 8.0 software (RRID:SCR_007370). The number of LysoTracker puncti and DAPI-stained nuclei were quantified according to user manual guidelines and puncti/nucleus calculated. All image analysis and quantification were performed under blinded conditions.

## Lifespan analysis

Survival assays were performed on once mated female flies, reared at standard densities, and transferred to vials (10–25/vial). Flies were transferred to fresh food every 2–3 days and deaths scored on transferal. Replicates for lifespans shown in *Figures 4E* and *5H* are shown in *Figure 4—figure supplement 1* and *Figure 5—figure supplement 1*. Source data for completed lifespans is available in *Source data 1*.

## Peptide digestion for LC-MS/MS analysis

Fly tissues (50/sample) from five biological replicates were lysed in pre-heated (95°C) 6 M guanidine chloride, 10 mM TCEP, 40 mM CAA, 100 mM Tris pH 8.5 lysis buffer. Following shaking at 1400 rpm (95°C) tissues were sonicated for five cycles (Bioruptor plus). Lysis buffer was then diluted 11-fold in digestion buffer (25 mM Tris 8.5 pH, 10% acetonitrile) and vortexed. Overnight trypsin (Trypsin Gold, Promega) digestion was carried out at 37°C at a 1:50 trypsin to protein ratio. Samples were sonicated again for five cycles, and further digested with gentle agitation at 37°C for 4 hr using a 1:100 trypsin to protein ratio. Samples were then placed in a SpeedVac (5 min, 37°C) to remove acetonitrile. Peptides were desalted using SDB.XC StageTips (*Rappsilber et al., 2003*). Peptides were then eluted using (80% acetonitrile, 0.1% formic acid), dried in a SpeedVac (35 min, 29°C), and quantified via Nanodrop.

## LC-MS/MS analysis

Peptides were loaded on a 50 cm column with a 75 µm inner diameter, packed in-house with 1.8 µm C18 particles (Dr Maisch GmbH, Germany) and dried via SpeedVac (35 min, 29°C). Reversed phase chromatography was performed using the Thermo EASY-nLC 1000. Buffer A was 0.1% formic acid and buffer B, 80% acetonitrile in 0.1% formic acid. Peptides were separated using a segmented gradient from 3% to 20% buffer B for 85 min and from 20% to 40% buffer B for 55 min. The Q-Exactive was operated in the data-dependent mode with survey scans acquired at a resolution of 120,000; the resolution of the MS/MS scans was set to 15,000. Up to the 20 most abundant isotope patterns with charge $\geq 2$ and<7 were selected for HCD fragmentation with an isolation window of 1.5 Th and normalised collision energies of 27. The maximum ion injection times for the survey scan and the MS/MS scans were 50 and 100 ms, respectively, and the AGC target value for the MS and MS/MS

scan modes was set to 1E6 and 1E5, respectively. The MS AGC underfill ratio was set to 20% or higher. Sequenced peptides were put on a dynamic exclusion for 45 s.

## Protein identification and quantification

Protein identification was carried out using MaxQuant (*Cox and Mann, 2008*) version 1.5.0.4 (RRID: SCR_014485) using the integrated Andromeda search engine (*Cox et al., 2011*). The data were searched against the canonical and isoform, Swiss1Prot and TrEMBL, Uniprot sequences corresponding to *Drosophila melanogaster* (20,987 entries). The database was automatically complemented with sequences of contaminating proteins by MaxQuant. For peptide identification, cysteine carbamidomethylation was set as 'fixed' and methionine oxidation and protein N"fixed' and methionine oxidation athionine oxidation at as 'fixed' and methionine oxidation aequences of contaminating proteins by MaxQuantor cleavage after lysine and arginine, also when followed by proline, and up to two missed cleavages. The minimum number of peptides and razor peptides for protein identification was 1; the minimum number of unique peptides was 0. Protein and peptide identification was performed with FDR of 0.01. In order to transfer identifications to peptides not selected for fragmentation in the separate analyses, the option to peptides protein identification was 1; the minimum number of unique peptides was 0. Proteinow' of 20 min. Protein and peptide identifications were performed within, not across, tissue groups. Label-free quantification (LFQ) and normalisation was done using MaxQuant (*Cox et al., 2014*). The default parameters for LFQ were used, except that the 'LFQ min. ratio count' parameter was set to 1. Unique plus razor peptides were used for protein quantification. LFQ analysis was done separately on each tissue.

## Perseus informatics analysis

The results of the LFQ analyses were loaded into the Perseus statistical framework (http://www.perseus-framework.org/) version 1.4.1.2 (RRID:SCR_015753). Protein contaminants, reverse database identifications and proteins 'Only identified by site' were removed, and LFQ intensity values were log2 transformed. After categorical annotation into four categories based on genotype (*dilp2-3,5/ wDah*) and presence of *Wolbachia* (yes/no), the data were filtered in order to contain a minimum of four valid values in at least one category. The remaining missing values were replaced, separately for each column, from normal distribution using width of 0.3 and down shift of 1.8.

## Proteomics differential expression analysis

We used the limma R package (version 3.30.13) to test the significance of the insulin response, that is the difference in protein expression between *dilp2-3,5* and wDah, in each of the four tissues (fat, brain, gut, thorax) and in the presence and absence of the endosymbiont *Wolbachia*. In addition, we determined the significance of the two-way interactions between *dilp2-3,5* and *Wolbachia* for each protein (*dilp2-3,5* - wDah vs. *dilp2-3,5*T - wDahT). p-Values were corrected for multiplicity on a per-tissue basis using the Benjamini-Hochberg method. *Wolbachia*-dependent proteins were defined as proteins with a significant change both between *dilp2-3,5* and wDah and in the two-way interaction between *Wolbachia*-status and insulin response (adj. p-value<=0.1). To identify *Wolbachia*-independently regulated proteins, we determined for each insulin-responsive protein whether it was equivalently regulated by *dilp2-3,5* and *dilp2-3,5*T, that is whether the 90% confidence interval of its log-fold-change fell within an interval ([−t; t], t = 0.085), by applying a TOST equivalence test on the fitted limma contrast. The threshold t was calculated by first calculating the median of interaction log2-fold changes in each tissue and then averaging them.

## RNAseq analysis

Isolation of RNA from frozen tissue samples of all genotypes (wDah control and *dilp2-3,5* flies, both in the presences and absence of *Wolbachia*) was performed on three independent biological replicates using Trizol Reagent (Thermo Fisher Scientific), according to the manufacturer's instructions, followed by DNase treatment (Qiagen). RNA quality was determined using the BioRad Experion (BioRad). RNA-seq library preparation and sequencing was performed by the Max Planck Genome centre Cologne, Germany (http://mpgc.mpipz.mpg.de/home). Stranded TruSeq RNA-seq library preparation was performed on 2 µg of total RNA after rRNA depletion (Ribo-zero). Brain samples were treated as above, however total RNA input was reduced (500 ng).

CircRNA data, extracted from the tissue-specific RNAseq data for *Wolbachia* positive *wDah* and *dilp2-3,5* mutants, have been previously published (*Weigelt et al., 2020*).

Libraries were sequenced with 37 mio, 100 bp single reads on an Illumina HiSeq2500 (Illumina). Adapter trimming was carried out using flexbar (version 2.5, RRID:SCR_013001), using minimum read length after trimming of 30, and quality threshold of 20. Transcripts were then mapped to the BDGP6.28 reference genome using tophat2 (version 2.1.0, RRID:SCR_013035) and counted via summarizeOverlaps (part of the Bioconductor R package GenomicAlignments, version 1.10.1, RRID: SCR_006442) with the option 'intersectionNotEmpty'. FPKM for comparison with the proteome were calculated using DESeq2 (version 1.14.1, RRID:SCR_015687). Differential expression analysis was carried out tissue-wise using a two-factor linear model with interaction effect in DESeq2, testing the insulin response in each tissue analogously to our proteomics analysis. p-Values were adjusted for multiplicity by DESeq2 using the Benjamini-Hochberg procedure on a per-tissue basis, with independent filtering enabled. *Wolbachia*-dependent transcripts were defined as transcripts with a significant change both between *dilp2-3,5* and wDah and in the two-way interaction between *Wolbachia*-status and insulin response (adj. p-value<=0.1). Due to lower power of the RNAseq assay compared to our proteomics, no *Wolbachia*-independently regulated transcripts were identified.

To determine transposon expression, we separately mapped all sample reads to canonical transposon reference sequences obtained from flybase (*Gramates et al., 2017*) using RSEM (RRID:SCR_013027). To get more robust estimates, dispersion and library normalisation factors were calculated based on the combined counts of both gene and transposon quantifications. We then fitted a two-factor linear model using edgeR (RRID:SCR_012802). Differential expression analysis was carried out as described for the primary analysis, but using the edgeR likelihood ratio test. p-Values were corrected for multiplicity using the Benjamini-Hochberg method. *Wolbachia*-dependent transposons were defined as transposons with a significant change both between *dilp2-3,5* and wDah and in the two-way interaction between *Wolbachia*-status and insulin *response* (adj. p-value<=0.1).

## Network propagation

We mapped the absolute log-fold changes of *dilp2-3,5* vs. wDah, as well as mNSC-ablated vs. wDah (*Tain et al., 2017*) comparisons to the protein-protein-interaction network obtained from DroID (RRID:SCR_006634) (*Murali et al., 2011*). Separately for both comparisons, we then performed network propagation (*Vanunu et al., 2010*), iteratively propagating the fold-change information through the network (spreading coefficient = 0.8) until a steady state was reached. In an additional step, we corrected for a known bias in network propagation that favours hub nodes by subtracting the propagation scores of in silico randomised fold changes from the initial result. We further integrated the scores from both models in each tissue into a consensus score. This was done by taking the minimum of both scores for each protein, effectively selecting proteins that were high-scoring in both. Next, we clustered network propagation scores across all tissues using hierarchical clustering in R (Ward's method). For each cluster, we carried out GO enrichment analysis (see 5.8) against the background set of all genes included in the network propagation analysis.

## GO enrichment analysis

Gene ontology annotations of genes and proteins were taken from the org.Dm.eg.db R package (version 3.4.0, Bioconductor RRID:SCR_006442). We excluded GO terms with fewer than five or more than 1000 annotated genes. For each gene list, significance of functional enrichment of GO terms in the gene list compared to the background list was determined using Fisher tests. For proteomics gene lists, the background was restricted to genes whose proteins were detected in the respective tissue. For transcriptomics gene lists, the background was restricted to genes whose transcripts were detected at a median count above five in the respective tissue. To reduce redundancy of functional categories, we clustered together GO terms that differed in five or fewer gene list member genes; the smallest GO term by total annotations was selected as the primary (most specific) term to represent the cluster. We then carried out correction for multiple hypothesis testing using the Benjamini-Hochberg method on all primary GO terms to obtain adjusted Fisher test p-values.

## Acknowledgements

We acknowledge contributions from the Proteomic facility at the Max Planck Institute for Biology of Ageing, the Max Planck Genome Center Cologne, the *Drosophila* transgenic facility at the Max Planck Institute for Biology of Ageing and the FACS and Imaging core facility of Max Planck Institute for Biology of Aging. We acknowledge funding from the Max Planck Society (to LT, RS, JP, RM, SG, JF, NN, MM, LP) and a Bundesministerium für Bildung und Forschung Grant SyBACol 0315893A-B (to AB, MC, LT, LP). The research leading to these results has received funding from the European Research Council under the European Union's Seventh Framework Programme (FP7/2007-2013) / ERC grant agreement number 268739 (to LP).

## Additional information

### Funding

| Funder | Grant reference number | Author |
|---|---|---|
| Max-Planck-Gesellschaft | | Luke Stephen Tain<br>Robert Sehlke<br>Ralf Leslie Meilenbrock<br>Thomas Leech<br>Jonathan Paulitz<br>Sebastian Grönke<br>Jenny Fröhlich<br>Matthias Mann<br>Nagarjuna Nagaraj |
| Bundesministerium für Bildung und Forschung | 0315893A-B | Luke Stephen Tain<br>Robert Sehlke<br>Manopriya Chokkalingam<br>Andreas Beyer |
| FP7 Ideas: European Research Council | FP7/2007-2013 | Linda Partridge |
| European Research Council | 268739 | Linda Partridge |

The funders had no role in study design, data collection and interpretation, or the decision to submit the work for publication.

### Author contributions

Luke Stephen Tain, Conceptualization, Resources, Data curation, Formal analysis, Supervision, Validation, Investigation, Visualization, Methodology, Writing - original draft, Project administration, Writing - review and editing; Robert Sehlke, Formal analysis, Investigation, Visualization, Methodology, Writing - original draft, Project administration; Ralf Leslie Meilenbrock, Jonathan Paulitz, Manopriya Chokkalingam, Formal analysis, Investigation, Methodology; Thomas Leech, Jenny Fröhlich, Investigation; Nagarjuna Nagaraj, Formal analysis, Methodology; Sebastian Grönke, Resources, Supervision; Ilian Atanassov, Formal analysis; Matthias Mann, Supervision; Andreas Beyer, Conceptualization, Formal analysis, Supervision, Funding acquisition, Writing - original draft, Writing - review and editing; Linda Partridge, Conceptualization, Supervision, Funding acquisition, Writing - original draft, Project administration, Writing - review and editing

### Author ORCIDs

Luke Stephen Tain ![ORCID] https://orcid.org/0000-0002-5845-7383
Robert Sehlke ![ORCID] https://orcid.org/0000-0002-6781-7272
Ralf Leslie Meilenbrock ![ORCID] http://orcid.org/0000-0001-9864-0257
Sebastian Grönke ![ORCID] http://orcid.org/0000-0002-1539-5346
Ilian Atanassov ![ORCID] http://orcid.org/0000-0001-8259-2545
Matthias Mann ![ORCID] http://orcid.org/0000-0003-1292-4799
Andreas Beyer ![ORCID] https://orcid.org/0000-0002-3891-2123
Linda Partridge ![ORCID] https://orcid.org/0000-0001-9615-0094

**Decision letter and Author response**
Decision letter https://doi.org/10.7554/eLife.67275.sa1
Author response https://doi.org/10.7554/eLife.67275.sa2

## Additional files

**Supplementary files**

• Source data 1. Raw lifespan data.

• Supplementary file 1. Total number of tissue-specific transcripts quantified and all tissue-specific changes in transcript expression between *dilp2-3,5* mutants and control (*wDah*) flies. Columns listed as: Tissue, Flybase ID, Gene name, base mean (Average transcript expression of the corresponding transcript in the respective tissue), LogFC p-value (Transcript log2(fold-change), *dilp*2-3,5 vs. *wDah*), *Wolbachia* minus adj p-value (Transcript p-value, *dilp*2-3,5/Wol- vs. *wDah*/Wol-), *Wolbachia* minus LogFC (Transcript FDR, *dilp*2-3,5/Wol- vs. *wDah*/Wol-). Wol+/- denotes the presence (+) or absence (-) of *Wolbachia* in formulae.

• Supplementary file 2. Total number of tissue-specific proteins quantified and all tissue-specific changes in protein expression between *dilp2-3,5* mutants and control (*wDah*) flies. Columns listed as: Tissue (Thorax, gut, brain or fat body), Uniprot ID, Flybase ID, Symbol (Gene), Entrez ID, Protein *Wolbachia* dependent expression (True/False), Protein *Wolbachia* independent expression (True/False), protein average expression (Label-free quantification intensity), protein *dilp* logFC (*dilp2-3,5 v. wDah*), protein *dilp* p-value (*dilp2-3,5 v. wDah*), protein *dilp* FDR (*dilp2-3,5 v. wDah*), Protein interaction p-value, Protein interaction FDR, RNA interaction p-value, protein TOST. Genotypes denoted as *wDah* and *dilp (2-3,5)* with *Wolbachia* and *wDahT* and *dilpT (2-3,5)* without *Wolbachia*.

• Supplementary file 3. Significant differentially expressed (*dilp2-3,5* vs *wDah*) transcript/protein pairs from fat, gut, brain, or thorax, in one of four quadrants shown in *Figure 1* in the presence of *Wolbachia* and Appendix 2 in the absence of *Wolbachia*. Columns listed as: Tissue, Quadrant (Quadrants by direction of log2(fold-change): I (protein+;transcript+), II (protein−;transcript+), III (protein−;transcript−), and IV (protein+;transcript−)), Protein flybase ID, protein symbol (Gene name), protein entrez ID, protein avg expr (Average LFQ intensity of this protein in all samples in the respective tissue), RNA avg expr (Average transcript expression of the corresponding transcript in the respective tissue), protein p-value (*dilp2-3,5 v. wDah*), RNA p-value (*dilp2-3,5 v. wDah*), Protein adj p-value (*dilp2-3,5 v. wDah* FDR), Protein logFC (*dilp2-3,5 v. wDah*), RNA LogFC (*dilp2-3,5 v. wDah*), opposite (Are protein and transcript changed in opposite direction?), *Wolbachia* dependence (Does the protein change in *dilp*2-3,5 vs *wDah* depend on *Wolbachia* status?), *wolbachia* independence (Is the protein change in *dilp*2-3,5 vs *wDah* independent of *Wolbachia* status?), RNA *Wolbachia* dependence (Does the transcript change in *dilp*2-3,5 vs *wDah* depend on *Wolbachia* status?), interaction p-value (Protein p-value, interaction term (*dilp*2-3,5 vs. *wDah* ~ Wol+ vs. Wol-)), interaction adj p-value (Protein FDR, interaction term (*dilp*2-3,5 vs. *wDah* ~ Wol+ vs. Wol-)), *dilp*T p-value (Protein p-value, *dilp*2-3,5/Wol- vs. *wDah*/Wol-), *dilp*T adj p-value (Protein FDR, *dilp*2-3,5/Wol- vs. *wDah*/Wol-), protein LogFC (Protein log2(fold-change), *dilp*2-3,5/Wol- vs. *wDah*/Wol-), TOST p-value (Protein p-value, two-one-sided-t-test of equivalence, interaction term (*dilp*2-3,5 vs. *wDah* ~ Wol+ vs. Wol-)), TOST adj p-value (Protein p-value, two-one-sided-t-test of equivalence, interaction term (*dilp*2-3,5 vs. *wDah* ~ Wol+ vs. Wol-)). Wol+/- denotes the presence (+) or absence (-) of *Wolbachia* in formulae. Genotypes denoted as *wDah* and *dilp (2-3,5)* with *Wolbachia* and *wDahT* and *dilpT (2-3,5)* without *Wolbachia*.

• Supplementary file 4. Significant differentially expressed (*dilp2-3,5* vs *wDah*) **protein**/transcript pairs from fat, gut, brain, or thorax, in one of four quadrants shown in *Figure 1* in the presence of *Wolbachia* and Appendix 2 in the absence of *Wolbachia*. Columns listed as: Tissue, Quadrant (Quadrants by direction of log2(fold-change): I (protein+;transcript+), II (protein−;transcript+), III (protein−;transcript−), and IV (protein+;transcript−)), Protein flybase ID, protein symbol (Gene name), protein entrez ID, protein avg expr (Average LFQ intensity of this protein in all samples in the respective tissue), RNA avg expr (Average transcript expression of the corresponding transcript in the respective tissue), protein p-value (*dilp2-3,5 v. wDah*), RNA p-value (*dilp2-3,5 v. wDah*), Protein adj p-value (*dilp2-3,5 v. wDah* FDR), Protein logFC (*dilp2-3,5 v. wDah*), RNA LogFC (*dilp2-3,5 v.*

*wDah*), opposite (Are protein and transcript changed in opposite direction?), *Wolbachia* dependence (Does the protein change in *dilp*2-3,5 vs *wDah* depend on *Wolbachia* status?), *Wolbachia* independence (Is the protein change in *dilp*2-3,5 vs *wDah* independent of *Wolbachia* status?), RNA *Wolbachia* dependence (Does the transcript change in *dilp*2-3,5 vs *wDah* depend on *Wolbachia* status?), interaction p-value (Protein p-value, interaction term (*dilp*2-3,5 vs. *wDah* ~ Wol+ vs. Wol-)), interaction adj p-value (Protein FDR, interaction term (*dilp*2-3,5 vs. *wDah* ~ Wol+ vs. Wol-)), *dilp*T p-value (Protein p-value, *dilp*2-3,5/Wol- vs. *wDah*/Wol-), *dilp*T adj p-value (Protein FDR, *dilp*2-3,5/Wol- vs. *wDah*/Wol-), protein LogFC (Protein log2(fold-change), *dilp*2-3,5/Wol- vs. *wDah*/Wol-), TOST P-value (Protein p-value, two-one-sided-t-test of equivalence, interaction term (*dilp*2-3,5 vs. *wDah* ~ Wol+ vs. Wol-)), TOST adj p-value (Protein p-value, two-one-sided-t-test of equivalence, interaction term (*dilp*2-3,5 vs. *wDah* ~ Wol+ vs. Wol-)). Wol+/- denotes the presence (+) or absence (-) of *Wolbachia* in formulae. Genotypes denoted as *wDah* and *dilp (2-3,5)* with *Wolbachia* and *wDahT* and *dilpT* (*2-3,5*) without *Wolbachia*.

• Supplementary file 5. GO enrichment analysis of significant differentially expressed (*dilp*2-3,5 vs *wDah*) protein/transcript pairs from fat, gut, brain, or thorax, in one of four quadrants shown in *Figure 1* in the presence of *Wolbachia*. Columns listed as: Tissue, Quadrant (Quadrants by direction of log2(fold-change): I (protein+;transcript+), II (protein−;transcript+), III (protein−;transcript−), and IV (protein+;transcript−)), Protein flybase ID, protein symbol (Gene name), protein entrez ID, protein avg expr (Average LFQ intensity of this protein in all samples in the respective tissue), RNA avg expr (Average transcript expression of the corresponding transcript in the respective tissue), protein p-value (*dilp2-3,5 v. wDah*), RNA p-value (*dilp2-3,5 v. wDah*), Protein adj p-value (*dilp2-3,5 v. wDah* FDR), Protein logFC (*dilp2-3,5 v. wDah*), RNA LogFC (*dilp2-3,5 v. wDah*), opposite (Are protein and transcript changed in opposite direction?), *Wolbachia* dependence (Does the protein change in *dilp*2-3,5 vs *wDah* depend on *Wolbachia* status?), *Wolbachia* independence (Is the protein change in *dilp*2-3,5 vs *wDah* independent of *Wolbachia* status?), RNA *Wolbachia* dependence (Does the transcript change in *dilp*2-3,5 vs *wDah* depend on *Wolbachia* status?), interaction p-value (Protein p-value, interaction term (*dilp*2-3,5 vs. *wDah* ~ Wol+ vs. Wol-)), interaction adj p-value (Protein FDR, interaction term (*dilp*2-3,5 vs. *wDah* ~ Wol+ vs. Wol-)), *dilp*T p-value (Protein p-value, *dilp*2-3,5/Wol-vs. *wDah*/Wol-), *dilp*T adj p-value (Protein FDR, *dilp*2-3,5/Wol- vs. *wDah*/Wol-), protein LogFC (Protein log2(fold-change), *dilp*2-3,5/Wol- vs. *wDah*/Wol-), TOST p-value (Protein P-value, two-one-sided-t-test of equivalence, interaction term (*dilp*2-3,5 vs. *wDah* ~ Wol+ vs. Wol-)), TOST adj p-value (Protein p-value, two-one-sided-t-test of equivalence, interaction term (*dilp*2-3,5 vs. *wDah* ~ Wol+ vs. Wol-)). Wol+/- denotes the presence (+) or absence (-) of *Wolbachia* in formulae. Genotypes denoted as *wDah* and *dilp (2-3,5)* with *Wolbachia* and *wDahT* and *dilpT* (*2-3,5*) without *Wolbachia*.

• Supplementary file 6. GO enrichment analysis of significant differentially expressed (*dilp2-3,5* vs *wDah Wolbachia*-dependent changes) proteins and transcripts. Each sheet shows tissue-specific (fat, gut, brain, or thorax) enrichments.

• Supplementary file 7. Network propagation scores and cluster assignments (*Figure 2*). Columns listed as: Flybase ID, Symbol (Gene), Cluster (related to *Figure 2*), abl (regulated in the mNSC-ablation model [*Tain et al., 2017*]), *dilp* (regulated in *dilp2-3,5 v wDah*), int (integrated responses between regulation in mNSC-ablation and *dilp2-3,5* models).

• Supplementary file 8. GO enrichment analysis results of network propagation clusters (*Figure 2*). Columns listed as: Cluster (Relates to *Figure 2*), Terms in Cluster (GO Terms significantly Enriched in a specific cluster), GO ID, Term, Ontology (BP, MF, CC), Annoatated (#genes annotated with the term in the tissue-specific background), Significant (#genes significant annotated with the term in the tissue-specific background), Expected (expected # of significant genes with the term in the tissue-specific background), Enrichment (Enrichment of the observed over the expected number of significant genes), Log2 of Enrichment, Fisher (p-value of enrichment), Primary Fisher adj (adjusted p-value of enrichment), Genes (Names of significant genes annotated with the term), Genes in secondary.

• Supplementary file 9. Log-fold changes of significant differentially expressed (*dilp2-3,5* vs *wDah*) protein associated with the regulation of DNA replication, DNA damage/ repair responses and related functions. LogFC, p-value, Fisher adj (adjusted P-value), Flybase ID, and Gene symbol (Name) are shown.

- Supplementary file 10. Analysis of transposon differential expression in *dilp2-3,5* flies (with/without Wolbachia) vs. *wDah*. Columns listed in summary tab.

- Transparent reporting form

## Data availability

Sequencing data have been deposited in GEO under accession code GSE122190 The mass spectrometry proteomics data have been deposited to the ProteomeXchange Consortium via the PRIDE partner repository with the dataset identifier PXD011589.

The following datasets were generated:

| Author(s) | Year | Dataset title | Dataset URL | Database and Identifier |
|---|---|---|---|---|
| Tain LS, Sehlke R, Meilenbrock RL, Leech T, Paulitz J, Chokkalingam M, Nagaraj N, Grönke S, Fröhlich J, Atanassov I, Mann M, Beyer A, Partridge L | 2021 | Tissue-specific modulation of gene expression in response to lowered insulin signalling in *Drosophila* | https://www.ncbi.nlm.nih.gov/geo/query/acc.cgi?acc=GSE122190 | NCBI Gene Expression Omnibus, GSE122190 |
| Tain LS, Sehlke R, Meilenbrock RL, Leech T, Paulitz J, Chokkalingam M, Nagaraj N, Grönke S, Fröhlich J, Atanassov I, Mann M, Beyer A, Partridge L | 2021 | Tissue-specific modulation of gene expression in response to lowered insulin signalling in *Drosophila* | https://www.ebi.ac.uk/pride/archive/projects/PXD011589/private | PRIDE, PXD011589 |

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

## Appendix 1

## Tissue-specific differential expression of the *Drosophila* proteome in response to reduced IIS at the level of proteome and transcriptome

(A)Pairwise correlations between replicates belonging to the same genotype, across all genotypes and tissues, for both transcriptomic and proteomic measurements. (B) Principle component analysis projections of all transcriptomic and proteomic replicates, respectively, showing clear separation of the tissues. 12 samples, comprising three biological replicates of the four genotypes (*wDah* and *dilp2-3,5* mutant flies plus/minus *Wolbachia*) are plotted for each tissue. For the proteomic analysis 20 samples, comprising five biological replicates of the four genotypes (*wDah* and *dilp2-3,5* mutant flies plus/minus *Wolbachia*) are plotted for each tissue. Proteomic replicates show relatively little variation and thus overlay each other. (C) Total number of transcripts and proteins (incl. isoforms mapped to unique flybase identifiers) that were detected across fat body, gut, brain, and thorax, and the overlap between tissues. Total numbers detected in each tissue are given in parentheses. (D) Number of differentially expressed transcripts and proteins (including isoforms) by tissue between *dilp2-3,5* mutants and controls (*wDah*) (adj. p-value≤0.1) across all tissues, as well as their overlap. Total numbers detected in each tissue are given in parentheses.

## Appendix 2

The Wolbachia-dependent transcriptome and proteome response to IIS. (A) Bioinformatics flowchart showing the derivation and numbers of transcripts/proteins that were differentially regulated in response to reduced IIS (*dilp2-3,5 vs wDah*), showed an interaction between the responses to reduce IIS and the presence of Wolbachia (*dilp2-3,5 - wDah vs. dilp2-3,5T - wDahT* (*T* denotes *Wolbachia* genotypes, see Methods section)) and if those changes in expression occurred in a Wolbachia-dependent or -independent manner in a tissue. Specific bioinformatics analyses are described in the methods section. (B) Numbers of transcripts and proteins in each tissue that were differentially regulated between *dilp2-3,5* and *wDah* (middle column), and at the same time detected as Wolbachia-independent (left column) or Wolbachia-dependent (right column), by tissue. (C) Significantly enriched (Fisher-test, FDR $\leq$ 0.05) functional GO terms among Wolbachia-dependent regulated transcripts and proteins in *dilp2-3,5* mutants, by tissue. Barcharts are segmented into cells that correspond to individual significantly regulated proteins. Colour indicates log-fold-change. Gene ontologies associated with DNA replication (*) and DNA damage (**). (D-E) Proportions of protein/transcript pairs that were concordantly regulated (both up or down) or oppositely regulated, in the absence of Wolbachia (*dilp2-3,5T vs. wDahT*). Correlations were calculated between the protein and transcript log-fold changes of significantly regulated protein/transcript pairs in each plot. (D) All protein/transcript pairs in the respective tissue where the transcript was independently significantly regulated (adj. p-value$\leq$0.1). (E) All protein/transcript pairs in the respective tissue where the protein was independently significantly regulated (adj. p-value$\leq$0.1).

## Appendix 3

Correlation and comparison of tissue-specific proteome remodelling between two independent models of reduced IIS. (A) Tissue-specific correlation of directional log-fold changes between *dilp2-3,5* mutants and controls (*wDah*) and mNSC-ablated flies and controls (*wDah*). Black dots show proteins with a significant combined P-value after multiplicity adjustment ($p \leq 0.1$). Numbers in quadrants show the number of these significant proteins in each quadrant. Circle-segments and percentages represent the fraction of significant proteins over total proteins in the given tissue. Red dots signify proteins that were individually significant in both proteomics comparisons. (B) Total number of differentially expressed proteins (including isoforms), irrespective of tissue, and tissue-specifically, in mNSC-ablated flies (*Tain et al., 2017*) and *dilp2-3,5* mutants (adj. p-value$\leq$0.1).

