## [Decision Letter]

**Acceptance summary:**

This elegant and thorough study enhances understanding of mechanisms underlying how reduced insulin signaling delays aging in different tissues. This paper also demonstrates the power of using tissue-specific profiling of transcriptome and proteome in combination with *Drosophila* molecular genetics for research to elucidate underlying mechanisms for health and longevity. The authors did an excellent job of addressing all three reviewers' concerns, and the reviewers are satisfied with the revision.

---

## [Author Response]

Reviewer #1 (Evidence, reproducibility and clarity (Required)):SummaryTain describes the impact of dilp2,3,5 mutant on transcriptional and protein profiles from fat body, gut, brain and thorax. This expands from previous work where profiles were generated from whole adults or when using different manipulations to alter insulin/IGF signaling. Transcriptional change correlated with protein profile change in most tissue, except fat body. These expression effects interacted with antibiotic treatment, which the authors attribute to Wolbachia; previous work showed that dilp2,3,5 only extends lifespan in antibiotic treated adults. From analysis of the expression changes, Tain tested candidate pathways for ability to modulate aging. Longevity could be increased by minichromosome gene expression in fat body, and lysosomal mannosidase gene expression in the gut. These results refine the catalog of potential cellular mechanisms by which altered IIS controls lifespan. The paper and conclusions should be improved at several points.Major commentsThe interaction with Wolbachia is interesting and important. The data imply there is some sort of age-associated damage caused by microbes that is forestalled by dilp235 mutation. When flies are treated with antibiotics, this damage absent and reduced IIS cannot improve survival. (This interpretation implies antibiotic treatment increases lifespan of wildtype flies but data among labs varies and no data on this point are given in the current paper.) The key problem for now is to show the effects are caused by Wolbachia rather than a general impact upon the fly microbiome. Tetracycline knocks down many bacteria and no evidence is given to distinguish that alternative from just removing Wolbachia. This issue must be experimentally resolved, or the work should be rewritten without claiming Wolbachia is the cause. I would also like to see data confirming Wolbachia is present and absent in the stocks currently used in the study (not from years past). And, when were the T+/- stocks produced, in this study or many years back (this matters because it affects time for other microbiome to recover)?

We thank the reviewer for identifying a clear misunderstanding with our methods. There was no treatment of flies with antibiotics during the experiments described in the manuscript. The description of the methods is now more transparent. For the generation of *Wolbachia* minus *wDah* stocks (wDahT), which naturally carry *Wolbachia*, were treated with tetracycline (25 mg/ml) for 3 generations. This was then followed by several generations to allow for full recovery from tetracycline treatment and to restore intestinal flora. Importantly, treatment with tetracycline and the resultant removal of *Wolbachia* did not alter lifespan in the *wDah* stocks [24, 27]. *dilp2,3-5* mutants, both with and without *Wolbachia,* were previously generated by Grönke et al. 2010. To achieve this, *dilp2-3,5* mutants were backcrossed for at least 10 generations into the previously described *wDah* or wDahT stock. The presence of *Wolbachia* was then checked by PCR as described above (Technical details are described in the updated methods).

We hope that this has clarified our methods to the reviewer. We have now expanded and clarified the methodology section text to better describe the generation of our *Wolbachia* positive and negative stocks. These changes are outlined below.

Original methods text in the “Fly stocks, fly husbandry, and dissection” section of the method section on pg.31

“All flies were backcrossed into a white Dahomey (*wDah*) strain genetic background for at least 8 generations. Wolbachia minus (wDahT) flies were generated as previously described [24], *wDah* flies were treated once with Tetracycline to remove Wolbachia.”

This section of the methodology text has now been edited as seen below.

“All flies were backcrossed into a white Dahomey (*wDah*) strain genetic background for at least 8 generations. Naturally Dahomey carry the intracellular bacterium *Wolbachia pipientis*. Wolbachia minus (*wDahT*) flies were generated by treating *wDah* flies with Tetracycline (25 mg/ml in standard SYA food) for 3 generations followed by a minimum of 5 generations to allow for full recovery from tetracycline treatment and restoration of intestinal flora. Unless specifically stated all strains, including *FB-Gal4, NP1-Gal4, UAS-MCM6,* and *UAS-LManV*, were backcrossed into a *Wolbachia* positive white Dahomey (*wDah*) strain genetic background. The presence or absence of *Wolbachia* was tested in all stocks via PCR using primers for the *Wolabchia surface protein (wsp)*, specifically the primer combination wsp81F (AAAAATTAAACGCTACTCCA) and wsp691R (TGGTCCAATAAGTGATGAAGAAAC)*. dilp2-3,5*T mutants were generated by back crossing flies into the *wDahT* background for a minimum of 10 generations [24].”

In addition, we have performed an additional PCR on all stocks used to show *Wolbachia* infection status for the reviewers (see below). As stated in the methods all stocks were positive for *Wolbachia* (*wDah, dilp2-3,5, UAS-MCM6, FB-Gal4, UASLManV, NP1-Gal4*, unless explicitly stated (*wDahT, dilp2-3,5T*)

Reviewer response Figure 1. *Wolbachia* infection status of all stocks used within the study. Samples shown represent pooled (5 flies) DNA from the following genotypes; *wDah, wDahT, dilp2-3,5, dilp2-3,5T, UAS-MCM6, FB-Gal4, UAS-LManV, NP1-Gal4*, and a no DNA control. The presence or absence of *Wolbachia* was tested in all stocks via PCR using primers for the *Wolabchia surface protein (wsp)*, specifically the primer combination wsp81F (AAAAATTAAACGCTACTCCA) and wsp691R (TGGTCCAATAAGTGATGAAGAAAC). Lower panel shows a positive PCR control for *Rpl32*. Hyperladder 50bp (Bioline) was loaded to indicate product size.

MCM6 expression was strongly elevated in dilp235 mutant fat body when Wolbachia positive. Tain establishes experimentally that transgene expression of this gene reduces His-foci and TEs. It extends lifespan, which transcends the paper from a correlational project to causal inference. I think this needs to go a bit further. What was the Wolbachia state of the flies in the survival study? I think they should be Wolbachia positive; please clarify. I also recommend treating these flies to remove Wolbachia and remeasuring survival; if the logic of this paper is robust we expect the benefit of MCM6 upon survival to not show up. It is difficult to conclude genome stability/TEs regulation by dilp235 as a mechanism to increase survival dependent on Wolbachia when the transgene survival experiment with MCM6 does not include a Wolbachia treatment.

We agree with the reviewer that the *Wolbachia* status of the MCM6 survival assay, along with all genotypes, is important information for the reader and we have now included information on the *Wolbachia* status of all genotypes in the methods section (see below). Unless specifically stated all fly stocks used in this study carry *Wolbachia.* Furthermore, as stated in the previous response, PCR confirmation of *Wolbachia* status for all stocks is shown above.

Method section additional text added (pg.31)

Unless specifically stated all strains, including *FB-Gal4, NP1-Gal4, UAS-MCM6,* and *UAS-LManV*, were backcrossed into the *Wolbachia* positive white Dahomey (*wDah*) strain.

The reviewer suggests to overexpress MCM6 also in the *Wolbachia* negative background to show that the beneficial effect of fat body-specific MCM6 expression requires the presence of *Wolbachia* in the wild type background, independent of reduced IIS. We thank the reviewer for their comment. Our data show that fat body-specific overexpression of MCM6 is sufficient to extend lifespan, independently of reduced IIS activity. We agree that we cannot exclude that this effect is independent of *Wolbachia* as all overexpression work was performed in *Wolbachia* positive backgrounds. However, *Wolbachia* status has no influence on either the lifespan (Grönke et al. 2010 [24 in the main text]) or the level of DNA damage (pHis2Av punctae/cell in Figure 4B) of *wDah* flies. As such we would not expect differences in DNA damage or lifespan upon overexpression of MCM6.

I have concerns on the genetics of the survival data. For MCM6 there is as much of a survival difference between the two parental genotypes as there is between the F1 and the gal4 control. This looks like hybrid vigor and you cannot rule that out by 'eight generations of backcrossing'. I would like to see a second design, familiar to the authors: drive MCM6 in adult fat body using geneswitch. That ensures all cohorts are coisogenic and keeps the expression just in adults, where you want it. I have the same concerns for the lysosome survival study.

We thank the reviewer for their comments. We will address these points in the revision. Specifically we intend to extend our lifespan analysis to included tissue specific Geneswitch drivers in the at body (for the over expression of UAS-MCM6) and in the gut (for the overexpression of LManV).

Minor commentsI found the manuscript too long and wordy. The results contain discussion and conclusions that are repeated in the Discussion. Remove redundancy. Throughout the paper, be more telegraphic and direct, in particular consider using less of the first person narrative style. Describe the data and not the author's actions; this will make the paper more concise and readable.

We thank he reviewer for their comment. We will make every effort throughout to ensure the text more concise and clear for the reader.

Reviewer #1 (Significance (Required)):The work adds to a growing list of potential downstream effects by which reduced insulin signaling extends lifespan. The paper extends approaches already published by the lab but with specific tissues and a different genotype of reduced IIS. It develops additional network level analyses, measures of tissue performance, and new pathways to consider. It builds on how Wolbachia is a crucial variable in the fly aging model, and while we don't understand this mechanism the point is strongly reinforced in this paper.The audience will span workers in aging biology, cell and genome homeostatsis, insulin/IGF biology, and -omics analysis.Referees cross-commentingI see we agree overall the submission could move forward when it addresses its needs for additional data, and improves its presentation. I think the work overstates it significance but I still find value in its results. The conclusions should be made more robust, especially those where the candidate genes are thought to control lifespan; I am perplexed why the authors did not use Geneswitch as a genetic design since this is standard operating procedure for that group. And the replicate survival plots in the supplemental data are both incomplete (missing a genotype) and show a transient period effect in the controls. The Wolbachia angle is crucial to the explanations of the results, but the paper lacks transparency and robustness on the nature of the stocks and their infection status, and the transgene longevity experiments are made without considering this variable. I want the revision to address these key issues.

We thank the reviewers for their specific comments and cross-commenting summary. As detailed above we will address the reviewers concerns both here, and in the revision. Firstly, we have addressed the issue of the *Wolbachia* status of our stocks above. Secondly, we will perform additional lifespan analysis using Geneswitch drivers as a genetic design to increase the robustness of the analysis.

Reviewer #2 (Evidence, reproducibility and clarity (Required)):In this manuscript entitled "Tissue-specific modulation of gene expression in response to lowered insulin signaling in *Drosophila*", the authors extended our knowledge regarding the causality of insulin signaling mutations in longevity, by performing tissue specific transcriptomic and proteomic analysis using fruit flies. Specifically, the authors determined transcriptomic and proteomic changes in four fly tissues (gut, brain, thorax and fat body) caused by two insulin-reduced mutations, dilp2-3,5 and wDah, with or without gut bacteria Wolbachia pipientis. They showed that the majority of expression changes in fat body of the insulin signaling mutant flies were post-transcriptional and Wolbachia dependent. They then found that the induction of DNA damage and repair proteins in fat body was sufficient for longevity in insulin mutants, supported by bioinformatic analysis and lifespan data with fat body-specific overexpression of MCM6. Lastly, they showed that gut-specific overexpression of lysosomal mannosidase, which they identified from their bioinformatic analysis, was sufficient to increase autophagy, gut homeostasis, and consequently lifespan. Following are my concerns the authors should address.Major commentsThe major claim of this study is tissue-specific modulation of gene expression on longevity and other physiological changes. However, the key data of the papers are insufficient for this claim. Following are two experiments they need to perform to further support their main argument.1. In Figure 4, the authors identified fat body-specific upregulation of DNA damage/repair responses in dilp2-3,5 mutants by measuring p-His2Av puncta in fat body. The authors need to measure the p-His2Av puncta in other tissues.2. In Figure 5, the authors demonstrated that gut-specific overexpression of lysosomal α-mannosidase (LManV) increases gut health and lifespan in flies. The authors need to test whether overexpression of LManV in other tissues affects lifespan.

The reviewer states “The major claim of this study is tissue-specific modulation of gene expression on longevity and other physiological changes. However, the key data of the papers are insufficient for this claim.” Our response to both major comments point 1 and 2 follows below.

We thank the reviewer for their comment and agree that we have examined physiological changes only in tissues in which the expression changes were identified in response to reduced IIS. Our manuscript details the identification of tissue-specific changes in gene expression in responses to reduced IIS. We use our analysis of those tissue-specific changes in response to reduced IIS to direct the tissue-specific expression of, potentially lifespan modulating, individual genes in the specific tissue in which those expression changes occurred. Our approach is an unbiased approach directed by out expression profiling. In our tissue-specific expression profiling we identified gut-specific up-regulation of LManV and a fat body-specific up-regulation of MCM6 in long-lived IIS mutant flies. Both responses are novel responses. We then showed gut- and fat body-specific expression of LManV or MCM6, respectively, was sufficient to extend lifespan in otherwise wild type flies. As such, we believe our manuscript shows tissue-specific modulation of gene expression is sufficient to induce physiological changes (reduced DNA damage/increased genome stability and increased gut health) and extend lifespan.

Furthermore, if increased expression of LManV or MCM6 in other tissues were to extend lifespan, possibly through increasing autophagy or DNA damage responses in that tissue, it would be interesting, but would be the result of ectopic expression and not related to expression changes in response to reduced IIS nor IIS-mediated longevity. Importantly, significant expression of LManV and MCM6 was mostly limited to the gut and the fat body respectively.

Minor comments1. The authors postulated that "post-transcriptional changes" is equivalent to "proteomic changes". In fact, "transcriptomic changes" also reflect post-transcriptional changes including splicing and RNA decay. The authors need to avoid using posttranscriptional changes and proteomic changes interchangeably and use an exact definition.

We thank the reviewer for bringing this to our attention and will edited the text throughout to and clarity and more precisely define our meaning upon full revision.

2. The logical flow from Figure 2 to Figure 3-5 is relatively weak. The authors should strengthen their argument with better logical flow.

The reviewer suggests strengthening the logical flow from Figure 2 to Figure 3-5. Our manuscript characterizes tissue-specific transcriptomic and proteomic expression changes in response to reduced IIS. The changes in gene expression are outlined in Figure 1 and Figure S1 A-E. Using two complimentary analyses (see below) we focused on investigated proteomic changes/responses to reduced IIS that may be causal for longevity.

To focus on expression changes that may contribute to the long life of dilp2-3,5 mutants we examined expression changes that required *Wolbachia* (Figure S2). The presence of *Wolbachia* is required for lifespan extension of *dilp2-3,5* mutants, but not for other IIS-related phenotypes (Grönke et al. 2010). Therefore, *Wolbacha-*dependent changes to the transcriptome or proteome may be causal for the longevity of IIS mutants.

To identify robust tissue-specific gene expression changes in response to reduced IIS we also performed hierarchical clustering and GO enrichment analysis of significantly regulated proteins in two independent models of reduced IIS activity, namely *dilp2-3,5* mutants and *mNSC-ablated* flies (Tain et al. 2017). Importantly, *dilp2-3,5* mutant flies are longer lived than mNSC-ablated flies and any *dilp2-3,5*-specific proteomic changes may provide insight into additional mechanisms regulating longevity. We identified several conserved proteomic changes (and processes linked to those proteins) between the two models, several of which have been directly associated with longevity of IIS mutants. In addition we identified fat body-specific ER protein targeting, and gut-specific carbohydrate metabolism/α glucosidase activity, specifically mannose metabolism, robust expression changes in response to reduced IIS. Furthermore, we identified DNA damage and repair responses as a functional signature present only in the fat body of dilp2-3,5 mutants. Therefore, our bioinformatic analyses led us to focus on three tissue-specific functional signatures/processes that may contribute the longevity in IIS mutants; translation/ER transport and DNA damage/repair responses in the fat body, and Mannose metabolism in the gut.

We have now edited the text following from the description of Figure 2 to better explain the logical flow from Figure 2 to Figure 3-5. The original text and edited text are shown below.

Original text:

“Combining our tissue-specific transcriptomic, proteomic, and cross model proteomic analyses, of gene expression remodelling in response to reduced IIS led us to examine three main findings that were functional candidates for relevance for longevity. First, we investigated the targeting and translation of proteins to the ER in fat body in more detail. Second, we analysed the importance of fat bodyspecific DNA damage and repair responses. Finally, we analysed gut-specific mannose metabolism. “

Edited text to emphasize the logical flow:

Combining our tissue-specific transcriptomic, proteomic, *Wolbachia-*dependent regulation, and cross model proteomic analyses, of gene expression remodelling in response to reduced IIS led us to examine three main findings that were functional candidates for relevance for longevity. Firstly, we investigated the targeting and translation of proteins to the ER in fat body in more detail. These functional signatures were enriched in the fat body of dilp2-3,5 mutants and linked to proteins whose levels decreased in response to reduced IIS despite increased expression of the associated transcripts (Figure 1B, Table 5). Furthermore, translational and ER-targeting functional signatures were conserved between dilp2-3,5 mutants and mNSC-ablated flies (Figure 2, cluster 7 and 8). Second, we analysed the importance of fat body-specific DNA damage and repair responses, whose functional signatures were identified as both *Wolbachia* dependent changes in response to reduced IIS and only present in dilp2-3,5 mutants. Finally, we analysed gut-specific mannose metabolism, which we identified as both gut-specific *Wolbachia-*dependent changes in response to reduced IIS, and conserved between two models of reduced IIS. “

3. In Figure 2, heat maps of network propagation scores do not provide any information of protein change directions. Moreover, it is ambiguous which cluster is specific to particular experimental conditions. Devise another quantified visualization with direction changes.

The reviewer is correct, the tissue-specific heat map shown in Figure 2 does not show any information on the directionality of protein expression changes in response to reduced. All proteins included in Figure 2 are listed within the associated Supplemental Table (Table S8) and the tissue-specific directionality data for every significantly regulated protein in response to reduced IIS is present within Supplemental Table 2.

The importance of the data represented in Figure 2 is not directionality, but rather identifying proteins whose expression is regulated in one or both models of reduced IIS (dilp2-3,5 mutants and mNSC-ablated flies), and in which tissues those changes occur. We believe that including directionality information in such a data representation as Figure 2 may overcomplicate the figure and draw attention away from the main point of the figure. The figure was designed to highlight tissue-specific functional signatures (through GO enrichment analysis) occurring in one model of reduced IIS (dilp2-3,5 mutants or mNSC-ablated flies), or both (shown as the “integrated” column in Figure 2).

We do agree with the reviewer that in addition it is important to see that there are not major discrepancies between the two models in the direction of the regulation on the individual protein level. This information is currently contained with Supplemental Table 2 and 8. To make this more readily accessible to the reader we will generate an additional supplemental figure in the final revision that shows side-by-side the directional changes of each protein.

4. The authors should cite sources of analysis tools in Methods.

We thank the reviewer for pointing out the missing citations and we will remedy this in the final revision upon completion of our additional analysis.

5. The authors need to correct errors and provide more information for followings:1. On page 5, it is unclear which means the “half” in Figure S1E. Explain what corresponds to 22 proteins and 17 transcripts in Figure S1E. In Figure S1B, specify the number of replicates. Samples in proteomics do not seem to be replicates. In Figure S1E, avoid using “#transcripts” and “#proteins”.On page 5, it is unclear which means the “half” in Figure S1E.

The text has been edited to remove this reference. Figure S1E (and associated text if the legend of Figure S1) has also been removed as it represents duplicate information from that found in Figure S1C.

Edited text

Overall both the proteomic and transcriptomic responses to reduced IIS were highly tissue-specific, only 22 proteins and 17 transcripts showed altered expression in all four tissues (Figure S1E).

Explain what corresponds to 22 proteins and 17 transcripts in Figure S1E.

In total we identified a total of 11331 transcripts and 7234 proteins (Figure S1AC), of which 3683 transcripts and 3738 proteins showed significantly altered expression in at least one tissue of *dilp2-3,5* mutant flies. Of those differentially regulated genes only 17 were regulated on the transcript level and 22 on the protein level across all 4 tissues tested here.

In Figure S1B, specify the number of replicates. Samples in proteomics do not seem to be replicates.

Per tissue, and per genotype we have a total of three biological replicates of the RNA-Seq and 5 biological replicates for the proteomic analysis, as stated in the methods section (see below). Given that we have 4 genotypes (*wDah* plus/minus *Wolbachia* and *dilp2-3,5* mutants plus/minus *Wolbachia)* we have a total of 12 samples per tissue in the RNA-seq and 20 samples per tissue in the proteomic analysis. The PCA analysis was performed on all samples of tissue, irrespective of the genotype to show clear separation between tissues. As mentioned below in the legend of Figure S1 (see below) the variability between samples of a single tissue was so low in comparison to the differences between tissues that in the PCA the replicates overlay each other.

Pg. 25 – Figure legend S1

(B) Principle component analysis projections of all transcriptomic and proteomic replicates, respectively, showing clear separation of the tissues. Proteomic replicates overlay each other.

The text of the Figure S1 legend has been edited (below) to convey this information more clearly to the reader.

Figure S1. Tissue-specific differential expression of the *Drosophila* proteome in response to reduced IIS at the level of proteome and transcriptome.

(A) Pairwise correlations between replicates belonging to the same genotype, across all genotypes and tissues, for both transcriptomic and proteomic measurements. (B) Principle component analysis projections of all transcriptomic and proteomic replicates, respectively, showing clear separation of the tissues. 12 samples, comprising 3 biological replicates of the 4 genotypes (*wDah* and *dilp23,5* mutant flies plus/minus *Wolbachia*) are plotted for each tissue. For the proteomic analysis 20 samples, comprising 5 biological replicates of the 4 genotypes (*wDah* and *dilp2-3,5* mutant flies plus/minus *Wolbachia*) are plotted for each tissue. Proteomic replicates show relatively little variation and thus overlay each other.

Pg. 35 – RNAseq analysis

Isolation of RNA from frozen tissue samples of all genotypes (*wDah* control and *dilp2-3,5* flies, both in the presences and absence of *Wolbachia*) was performed on three independent biological replicates

Pg. 33 – Peptide digestion for LC-MS/MS analysis

Fly tissues (50/sample) from five biological replicates were lysed in pre-heated (95oC) 6 M guanidine chloride, 10 mM TCEP, 40 mM CAA, 100 mM Tris pH 8.5 lysis buffer.

2) On page 5, please explain "mirrored" in detail. In Figure 1A, the sum of proportion in gut exceeds 100%. In Figure 1B, correlation coefficients between proteome and transcriptome are lower in brain and thorax than fat body. Nevertheless, the authors mentioned "gene expression changes in response to lowered IIS in the fat body were primarily post-transcriptional."

We thank the reviewer for their comment. Mirrored was used to describe concordant co-regulation of transcript and protein abundance in response to reduced IIS. We have now edited this text to ensure this is clear (see below).

On average across the four tissues, two thirds of the significant tissue specific changes in transcript levels in *dilp2-3,5* mutants were matched by concordant changes in expression of the encoded proteins (Figure 1A, Table 3).

The reviewer is correct the sum of the percentages exceeds 100%, by 1%. This is a by-product of summing rounded percentages. This is now specifically mentioned in the figure legend.

We have edited the text to state

However, many gene expression changes in response to lowered IIS in the fat body were post-transcriptional (Figure 1B).

3) On page 6, it is unclear which "729 proteins" were represented in Figure 1B. It is uncertain how many proteins are upregulated or downregulated.

The 729 proteins refers to 729 proteins, of the 1569 significantly regulated proteins in the fat body of *dilp2-3,5* mutants, whose expression is co-regulated with their associated transcripts. In Figure 1B the 729 proteins whose expression follows this pattern is shown in orange and blue, of which 26% were up regulated and 21% were down regulated.

The text has been edited for clarity (Pg.6).

Of those, 729 proteins changed expression in the same direction as their transcripts and were enriched for functions associated with proteostasis, amino acid metabolism, and mitochondria (Figure 1B shown in blue and orange, Table 5).

4) On page 7, it is difficult to understand what "high degree of post-transcriptional regulation" is.

Of those genes whose Wolbachia-dependent expression was altered in the fat body, only 26 were regulated on both the transcript and protein level. This concordant with our previous analysis in the fat body suggesting a high degree of post-transcriptional regulation in response to reduced IIS (Figure 1B).

Text has been edited to

Of those genes whose *Wolbachia*-dependent expression was altered in the fat body, only 26 were regulated, and regulated in the same direction, on both the transcript and protein level, suggesting considerable post-transcriptional regulation in response to reduced IIS, specifically in this tissue (Figure 1B).

The wording "entirely dependent" seems too strong.

Text edited to state “dependent” not “entirely dependent”

5) On page 10, the authors mentioned “Tellingly, several components of the ER import and co-translational targeting machinery were down-regulated in the fat bodies of the two IIS mutants (Figure 3B).” In fact, these proteins do not seem to be significantly downregulated upon mNSC-ablation in Figure 3B.

Text edited

Tellingly, several components of the ER import and co-translational targeting machinery were down-regulated in the fat bodies of DILP2-3,5 mutants, and showed a similar trend in mNSC-ablated flies (Figure 3B).

6. Please describe in Figure 1 legend in detail.

We thank the reviewer for their point and have now expanded the legend of Figure 1 to increase clarity for the reader.

Edited text Pg. 22

Figure 1. Reducing IIS modulates both the tissue-specific transcriptomic and proteomic landscapes. Plots show the proportion of protein/transcript pairs that are co-regulated regulated in the same (both up (orange) or both down (blue)) or opposite (grey) directions in response to reduced IIS (*dilp2-3,5* vs. *wDah*). Correlations were calculated between the protein and transcript log-fold changes of significantly regulated protein/transcript pairs in each plot. (A) All protein/transcript pairs in the respective tissue where the transcript is significantly regulated (adj. P-value ≤ 0.1) in response to reduced IIS, irrespective of if the associated protein is significantly regulated (Table 3-4). (B) All protein/transcript pairs in the respective tissue where the protein is significantly regulated (adj. P-value ≤ 0.1) in response to reduced IIS, irrespective of if the associated transcript was significantly regulated (Table 3-4). Correlations (cor.), number of protein/transcript pairs (n) shown above each plot. Percentages of protein/transcript pairs within a specific quadrant of the plots are shown within the respective quadrants.

7) In Figure 3B, describe categories in separated heat maps.

The separated heatmaps shown in Figure 3B show the proteomic log fold changes of all proteins directly linked to the SRP translocon. This is stated in the legend of Figure 3 (See below).

Figure 3. Tissue-specific regulation of ER associated cellular compartments and the ER co-translational targeting machinery in two independent models of reduced IIS. (A) Heatmap of mean log-fold-changes in proteins annotated with selected GO cellular compartment terms, in the contrasts dilp2-3,5 vs. *wDah* (dilp2-3,5), dilp2-3,5T vs. wDahT (dilp2-3,5T), and InsP3-Gal4/UASrpr vs. *wDah* (mNSC-abl.) flies [28]. Significance of difference vs. zero was calculated using a two-sided t-test (*p<0.05,**p<0.01,***p≤0.001). (B) Changes in protein expression of SRP, SRP-receptor (SRP-R) sub-units, TRAM and translocon components. Asterisks indicate BH-corrected significance of the limma moderated t-test (*p <0.1, **p< 0.01, ***P < 0.001.

8) In Figure 5A, there is no data in brain or thorax. In Figure 5 (not in the legend), add a description about white, yellow, and red circles.

We apologise for any confusion caused by the plots shown in Figure 5A. The plots show only log-fold change of lysosomal α-mannosidases whose expression was significantly regulated in response to reduced IIS (shown in the white circles). In the case of the lysosomal α-mannosidases we only detected significant regulation of the transcripts in the fat body and the gut (Figure 5A) and in the fat body, gut, and brain on the level of the proteins (Figure 5B). We have now included data on the detected, but not significantly regulated transcripts and proteins (shown in grey)

We have edited the figure legend to state that both significant and non-significant expression changes are shown. Furthermore, detection of lysosomal α mannosidase proteins was mostly limited to the gut and fat body. We have also added a colour key description.

Legend now edited as below.

Figure 5. Lysosomal α-mannosidase expression is gut-specifically increased in response to reduced IIS, and gut-specific expression is sufficient to maintain gut health and extend lifespan. (A) Significant (white circles) and non-significant (grey points) tissue-specific log2-fold change of lysosomal α-mannosidase transcript (A) and protein (B) expression in dilp2-3,5 vs. *wDah*. Significantly regulated proteins (white circles), Wolbachia-dependent (red circles) regulation (adj. P-value ≤ 0.1). Directional significance established by one sided Student’s ttest. (C) Quantification of LysoTracker Red stained vacuoles per nucleus in the gut of flies gut-specifically overexpressing LmanV (Np1-Gal4;UAS-LmanV, n = 12, Red circles) compared to genetic controls (Np1-Gal4/+, n = 8 , white circles and UASLManV/+, n = 9, orange circles). Chart shows mean and error bars represent S.E.M. Significance determined by Kruskall-Wallace test and Dunn’s multiple comparisons test. (D) Proportion of aged (65d) flies exhibiting gut barrier function failure in response to gut-specific overexpression of LmanV (Np1Gal4;UAS-LmanV red circles) compared to genetic controls (Np1-Gal4/+ white circles and UAS-LmanV/+ orange circles) (n = 10). Significance determined by Kruskal-Wallis test and Dunn’s multiple comparisons test. € Age-related changes in intestinal stem cell proliferation in response to gut-specific overexpression of LmanV (Np1-Gal4;UAS-LmanV, n = 10, 65d, red circles) compared to genetic controls (Np1-Gal4/+ white circles, n = 7, and UAS-LmanV/+ orange circles, n = 11). Significance determined by Kruskal-Wallis test and Dunn’s multiple comparisons test. (F) Representative images and (G) quantification of age-related dysplasia in gut epithelia. Gut-specific overexpression of LmanV (Np1Gal4;UAS-LmanV red circles) significantly reduced age-related intestinal dysplasia in 65-day-old flies compared to genetic controls (Np1-Gal4/+ white circles and UAS-LmanV/+ orange cirlces). Significance determined by One-way ANOVA and multiple comparisons (Sidak’s). (H) Survival analysis of flies gut-specifically overexpressing LmanV (Np1-Gal4;UAS-LmanV, red) compared to the UASLManV/+ (orange) and Np1-Gal4/+ (black) heterozygous controls. Statistical significance between survival curves was determined by log-rank test (n = 150).

** <0.01, *<0.05.

9) In Figure S2A, the explanation of the interaction is insufficient. The reason why classification of transcripts is not in the figure is not obvious. In Figure S2C, it is better to mark which GO terms are associated with DNA replication and DNA damage/repair responses, respectively. Use complete description of “oxidoreductase activity.…” In Figure S2 legend, explain what stacking is in a single bar in Figure S2C.

We thank the reviewer for their comment. To ensure it is clear for both the reviewer and the reader we have included a more detailed explanation of the interaction term in the figure legend and refer readers to the methods section for a more full description (see below).

The same logic was used to classify both proteins and transcripts as *Wolbachia* dependent or not. We initially included only protein data to show the bioinformatics scheme. We thank the reviewer for bringing this to our attention and have now edited Figure S2A to include numbers associated to transcript classification.

Figure S2C and its corresponding legend has been edited to highlight specifically the GO terms associated with DNA damage and DNA replication, see below.

Figure S2. The *Wolbachia*-dependent transcriptome and proteome response to IIS. (A) Bioinformatics flowchart showing the derivation and numbers of transcripts/proteins that were differentially regulated in response to reduced IIS (*dilp2-3,5 vs wDah*), showed an interaction between the responses to reduce IIS and the presence of *Wolbachia* (*dilp2-3,5 – wDah vs. dilp2-3,5T – wDahT*) *and if those changes in expression occurred in a Wolbachia*-dependent or -independent manner in a tissue. Specific bioinformatics analyses are described in the methods section. (B) Numbers of transcripts and proteins in each tissue that were differentially regulated between *dilp2-3,5* and *wDah* (middle column), and at the same time detected as *Wolbachia*-independent (left column) or *Wolbachia* dependent (right column), by tissue. (C) Significantly enriched (Fisher-test, FDR ≤ 0.05) functional GO terms among *Wolbachia*-dependent regulated transcripts and proteins in *dilp2-3,5* mutants, by tissue. Barcharts are segmented into cells that correspond to individual significantly regulated proteins. Colour indicates logfold-change. Gene ontologies associated with DNA replication (*) and DNA damage (**). (D-E) Proportions of protein/transcript pairs that were concordantly regulated (both up or down) or oppositely regulated, in the absence of Wolbachia (*dilp2-3,5T vs. wDahT*). Correlations were calculated between the protein and transcript log-fold changes of significantly regulated protein/transcript pairs in each plot. (D) All protein/transcript pairs in the respective tissue where the transcript was independently significantly regulated (adj. P-value ≤ 0.1). € All protein/transcript pairs in the respective tissue where the protein was independently significantly regulated (adj. P-value ≤ 0.1).

We thank the reviewer for bringing this to our attention and have included the full description of the shortened GO term which now is edited to:

oxidoreductase activity, acting on CH-OH group of donors

“In Figure S2 legend, explain what stacking is in a single bar in Figure S2C”

The stacking within a single bar of the bar charts shown in Figure S2C allows the reader to see the number and the log-fold change (in response to reduced IIS) of individual regulated proteins associated to a specific enriched GO term. This is stated in the figure legend.

“Barcharts are segmented into cells that correspond to individual significantly regulated proteins. Colour indicates log-fold-change”

10) In Figure S3B, calculate representation factor and significance of the overlaps.

This information will be added to Figure S3B in the final revision.

11) The authors wrote “p-value” in various ways, including P-value, p-value, p-value, pvalue, and p value in figure legends, supplemental figures, and supplemental tables. Choose one among P-value {less than or equal to} 0.1, p{less than or equal to}0.1, p {less than or equal to} 0.1, p <= 0.1, and <.1. Use [n=10] or [n = 10] not [n= 10].

This information has been standardised throughout.

12) Use one between “log-fold-changes” and “log-fold changes”. Use “logrank test or logrank test” instead of “Log Rank test”. Use “one sided Student’s t-test” instead of “Onesided ttest” or “Student’s ttest”. Use “one-way ANOVA” not “One-way ANOVA”. Use “(one or two sided) Fisher’s exact test” instead of “Fisher test”.

This information has been standardised throughout.

13) Use the full name of “BH-correction”.

This information has been standardised to Benjamini Hochberg throughout.

14) On page 6, use “gut, brain, and thorax” instead of “brain gut and thorax”.

Text corrected

15) In Figure 4D, align boxes with legends.

Corrected

16) In Figure S1A, use transcriptomics instead of RNAseq. In Figure S1B, use point boundaries in the PCA. In Figure S1C and D, choose between either -omics and -ome. In Figure S1 legend, specify that “incl.” means “including”.

Edited and corrected. In Figure S1D incl. is now spelt out as including.

17) In Figure S2B, use same colors in transcripts and proteins.

Changed.

18) In Figure S3B, use same colors in Venn diagrams.

Colour has now been added to the Venn diagrams of Figure S3B.

19) In Figure S4, use a colored rectangle instead of a colored line.

Changed.

20) On page 4, "increased expression" instead of "increase expression".

Text edited.

21) On page 6, use comma for separation and change "gut brain and thorax" to "gut, brain, and thorax".

Text edited.

22) On page 7, use italics for genotypes "dilp2-3,5". Use "GO enrichment analysis" not "GO enrichment".

Text edited.

23) On page 10, remove "regulated" from "a post-transcriptionally regulated increased abundance".

Text edited.

24) On page 13, use "average log-fold changes" not "average log-old changes".

Text edited.

25) On page 14 and 24, change "Lysosomal" to "lysosomal".

Text has been edited.

26) Add a space between number and unit. For example, on page 23, "5µM" should be changed to "5 µM".

Text has been edited.

27) On page 36, change "proteome" to "transcriptome".

This sentence refers to using FPKM values, calculated using DESeq2, to compare expression between the transcriptome and the proteome. In this instance proteome was used correctly.

28) In Figure 4C, change "-3" to "<-3".

Corrected.

29) In Figure 5 legend, B to F should be C to H.

Corrected.

30) In Figure S2B, change "# wolbachia−dependent transcripts" to "Number of Wolbachia−dependent transcripts". In Figure S2 legend, begin sentences with capital letters.

Corrected.

Reviewer #2 (Significance (Required)):This study will contribute to aging research by helping understanding how reduced insulin signaling delay aging in different tissues by employing various factors. This paper also demonstrated power of approach using tissue-specific profiling of transcriptome and proteome in combination with *Drosophila* molecular genetics for research to elucidate underlying mechanisms for health and longevity.My expertise is genetics of aging research.Referees cross-commentingI agree with the other two reviewers' comments. By addressing all three reviewers' comments, this nice paper will be improved substantially.Reviewer #3 (Evidence, reproducibility and clarity (Required)):In this manuscript, Tain and colleagues reported a comprehensive transcriptomic and proteomics analysis on *Drosophila* IIS mutants. They found that flies with reduced IIS signaling exhibited tissue-specific transcriptional and proteomic profiles. Interestingly, the transcriptional changes in fat body of IIS mutants were not always consistent with the proteomic changes. Through bioinformatics and functional characterization, the authors identified three enriched pathways that are potentially associated with IISregulated longevity, 1) Protein targeting and translation in ER of the fat body; 2) Fat body-specific DNA damage and repair responses; 3) Gut-specific mannose metabolism. Lastly, overexpression of either minichromosome maintenance protein subunit MCM6 in the fat body or lysosomal α-mannosidase (LManV) in the gut extends lifespan. The study is well designed and the manuscript is well prepared. The work will contribute significantly to our understating of the evolutionarily conserved longevity pathway like IIS. I only have a few minor concerns listed below that need to be addressed before the manuscript is acceptable for publication.1) To narrow down the candidate genes that may have a causal role in IIS-regulated longevity, the authors examined the Wolbachia-independent and Wolbachia-dependent changes. As an alternative, it would be great if the authors could compare their -omics data to previous aging transcriptomes to identify IIS-regulated candidate genes with age-related expression profiles.

We agree with the reviewer that comparing our –omics data to previously published aging/IIS responsive transcriptomic datasets would be very interesting, if those transcriptomic datasets were tissue-specific. Unfortunately, very few datasets that examine tissue-specific gene expression changes in response to reduced IIS exist. Furthermore, those datasets that do exist use different platforms, for example Microarrays (Alic et al. 2014) that result in noncomparable levels of sensitivity. Alic et al. 2014 used Dros.2 microarrays to determine differential gene expression in the fat body of flies overexpressing dFoxo under the control of the geneswitch driver S106. They detected 174 differentially expressed genes with an FDR of 10%. In our current study, using RNA-Seq, we detect differential expression of over a 1000 transcripts in the fat body of *dilp2-3,5* mutants, almost 600 of which were regulated in that tissue alone.

In addition to compatibility issues arising from analysis platforms, the tissue specific nature of the current and previous –omic profiling studies is very important in performing meta-analyses of the kind suggested by the reviewer. Our current analysis highlights the tissue-specific nature of both transcriptomic and proteomic responses to reduced IIS (Figure S1D). Only 17 transcripts and 22 proteins were regulated in all 4 tissues of the dilp2-3,5 mutants examined here (Figure S1D). The majority of differentially expressed transcripts and proteins show differential expression in one tissue alone. For example, of the 1004 significantly regulated transcripts in the fat body of *dilp2-3,5* mutants, 590 were regulated in the fat body alone.

Therefore, whilst we do believe the meta-analysis approach suggested by the reviewer would be beneficial to the field and would lead to the discovery of further prolongevity targets, the tissue-specific RNA-Seq datasets to do such meta analyses in flies do not yet exist.

2) The identification of ER co-translational targeting and protein translocation was very intriguing. However, no functional characterization was shown. Have the authors tried to examine the role of TRAM and SRP subunits in lifespan control?

We agree with the reviewer that altered ER co-translational targeting and protein translocation in response to reduced IIS is very interesting, and is a possible direction for future work. However, we have, as of yet, not examined the role of SRP subunits in the tissue-specific regulation of lifespan.

3) The rationale behind the analysis on transposable elements (TEs) was a little unclear. Is TE enriched in any of the GO analysis? And the authors did not provide evidence supporting a role of IIS mutant in regulating age-related changes in TE expression.

We thank the reviewer for their comment and address their points below

GO terms associated with TE expression was not specifically enriched in our analysis. We used TE expression analysis as a proxy for genomic stability, which along with DNA damage was enriched in our GO enrichment analysis, specifically in the fat body of long-lived *dilp2-3,5* mutants. As described in our main text (Pg. 12-13) genome stability declines with age, and is itself a hallmark of ageing ([46,47]). Loss of genome stability is associated with increased mis-expression of transposable elements. Thus, any reduction in TE expression, by increased genome stability, in response to reduced IIS may be a mechanism of IIS-mediated longevity.

We agree with the reviewer that all evidence associated to changes in TE expression in response to reduced IIS is limited to our single 10d age. As the changes we see in TE expression are likely due to global changes in genome stability and not of specific TEs, performing expression analyses on individual TEs is unlikely to yield a robust and conclusive answer to age-related changes in TE expression. Therefore we propose to examine global TE expression in IIS mutants utilising available *Drosophila* transcriptomic datasets. One such dataset was published recently in *Wolbachia* positive *dilp2-3,5* mutants (Weigelt et al. 2020 doi.org/10.1016/j.molcel.2020.06.011). We will perform this analysis upon full revision.

Reviewer #3 (Significance (Required)):This study revealed interesting tissue-specific responses to reduced IIS signaling, including many previously uncharacterized transcriptional and proteomic changes. The work will contribute significantly to our understating of the evolutionarily conserved longevity pathway like IIS.